# Interplay between bacterial deubiquitinase and ubiquitin E3 ligase regulates ubiquitin dynamics on Legionella phagosomes

Shuxin Liu[1†], Jiwei Luo[2,3†], Xiangkai Zhen[2,3], Jiazhang Qiu[4*], Songying Ouyang[2,3*], Zhao-Qing Luo[1,5*]

[1]Department of Respiratory Medicine and Center of Infection and Immunity, Key Laboratory of Organ Regeneration and Transplantation of the Ministry of Education, The First Hospital, Jilin University, Changchun, China; [2]The Key Laboratory of Innate Immune Biology of Fujian Province, Provincial University Key Laboratory of Cellular Stress Response and Metabolic Regulation, Biomedical Research Center of South China, Key Laboratory of OptoElectronic Science and Technology for Medicine of the Ministry of Education, College of Life Sciences, Fujian Normal University, Fuzhou, China; [3]Laboratory for Marine Biology and Biotechnology, Pilot National Laboratory for Marine Science and Technology (Qingdao), Qingdao, China; [4]Key Laboratory of Zoonosis, Ministry of Education, College of Veterinary Medicine, Jilin University, Changchun, China; [5]Department of Biological Sciences, Purdue University, West Lafayette, United States

**\*For correspondence:**
qiujz@jlu.edu.cn (JQ);
ouyangsy@fjnu.edu.cn (SO);
luoz@purdue.edu (Z-QL)

[†]These authors contributed equally to this work

**Competing interests:** The authors declare that no competing interests exist.

**Abstract** *Legionella pneumophila* extensively modulates the host ubiquitin network to create the Legionella-containing vacuole (LCV) for its replication. Many of its virulence factors function as ubiquitin ligases or deubiquitinases (DUBs). Here, we identify Lem27 as a DUB that displays a preference for diubiquitin formed by K6, K11, or K48. Lem27 is associated with the LCV where it regulates Rab10 ubiquitination in concert with SidC and SdcA, two bacterial E3 ubiquitin ligases. Structural analysis of the complex formed by an active fragment of Lem27 and the substrate-based suicide inhibitor ubiquitin-propargylamide (PA) reveals that it harbors a fold resembling those in the OTU1 DUB subfamily with a Cys-His catalytic dyad and that it recognizes ubiquitin via extensive hydrogen bonding at six contact sites. Our results establish Lem27 as a DUB that functions to regulate protein ubiquitination on *L. pneumophila* phagosomes by counteracting the activity of bacterial ubiquitin E3 ligases.

## Introduction

The Gram negative bacterium *Legionella pneumophila* is an opportunistic pathogen ubiquitously found in natural and man-made water systems, often by association with amoebae species (*Fields et al., 2002*). Infection of humans by *L. pneumophila* occurs when susceptible individuals inhale contaminated aerosols, which introduce the bacteria to the lungs where alveolar macrophages engulf them by phagocytosis (*Newton et al., 2010*). The bacterial phagosome called the Legionella-containing vacuole (LCV) initiates a trafficking route that bypasses the endocytic maturation pathway (*Xu and Luo, 2013*). Instead, it appears to intercept vesicles originating from the endoplasmic reticulum (ER) and is eventually converted into a compartment whose membranes resemble those of the ER (*Swanson and Isberg, 1995*; *Tilney et al., 2001*).

The conversion of the plasma membranes from nascent phagosomes into the LCV with ER properties largely is mediated by virulence factors delivered into host cells via the Dot/Icm type IV secretion system of *L. pneumophila* (*Isberg et al., 2009*). These virulence factors, also called effectors, interfere with such diverse cellular processes as membrane trafficking, autophagy, protein translation and immunity by diverse mechanisms (*Qiu and Luo, 2017a*). The modulation of these processes is mediated by targeting key regulatory proteins via effector-induced post-translational modifications, including phosphorylation (*Haenssler and Isberg, 2011*), methylation (*Rolando et al., 2013*), phosphorylcholination (*Mukherjee et al., 2011*; *Tan et al., 2011*), AMPylation (*Hardiman and Roy, 2014*; *Müller et al., 2010*; *Tan and Luo, 2011*), and ubiquitination (*Qiu and Luo, 2017b*) or by subverting the metabolism of lipids key in cell signaling (*Swart and Hilbi, 2020*), including the production of phosphatidylinositol-4-phosphate (PI4P) on the LCV (*Hsu et al., 2012*) and the removal of phosphatidylinositol-3-phosphate from distinct organelles by phospholipase (*Gaspar and Machner, 2014*) or PI phosphatases (*Toulabi et al., 2013*).

Modulation of the ubiquitin network has emerged as an important theme in interactions between *L. pneumophila* and its hosts. More than 10 Dot/Icm effectors have been found to function as ubiquitin E3 ligases via various mechanisms (*Qiu and Luo, 2017b*). Some of these proteins possess structural domains such as F-box and U-box found in components of mammalian E3 ligase complex (*Ensminger and Isberg, 2010*; *Habyarimana et al., 2008*; *Lin et al., 2015*; *Kubori et al., 2010*), whereas others harbor cryptic motifs that catalyze the ubiquitin ligation reaction by unique mechanisms (*Hsu et al., 2014*; *Lin et al., 2018*). Most unexpectedly, members of the SidE effector family catalyze phosphoribosyl ubiquitination on serine residues of several small Rab small GTPases and of the ER protein Rtn4 by a NAD-dependent mechanism that bypasses the host E1 and E2 enzymes (*Bhogaraju et al., 2016*; *Kotewicz et al., 2017*; *Qiu et al., 2016*). Interestingly, the activity of these unique ubiquitin ligases is regulated by bacterial enzymes that function either to reverse the ubiquitination (*Shin et al., 2020*; *Wan et al., 2019*) or to directly block the ubiquitin activation step of the catalysis by calmodulin-dependent glutamylation (*Bhogaraju et al., 2019*; *Black et al., 2019*; *Gan et al., 2019b*; *Sulpizio et al., 2019*).

Ubiquitination can be reversed by specific deubiquitinases (DUBs), which are a large group of proteases functioning to cleave ubiquitin moieties from modified substrates (*Komander and Rape, 2012*). The activity of DUBs restores ubiquitinated proteins back to their original forms thereby reversing the effects caused by ubiquitination. Coordinated activity of E3 ubiquitin ligases and DUBs thus controls the fate of modified substrates (*Popovic et al., 2014*; *Qiu and Luo, 2017b*). According to their structures and mechanisms of action, DUBs are classified into several subfamilies, including ubiquitin-specific proteases (USPs), ubiquitin C-terminal hydrolases (UCHs), ovarian tumor proteases (OTUs) (*Balakirev et al., 2003*), Machado-Joseph disease protein domain proteases (MJDS), JAMM/MPN domain-associated metallopeptidases(JAMMs) (*Ronau et al., 2016*) and the more recently identified MINDY and ZFUSP family of DUBs (*Abdul Rehman et al., 2016*; *Hermanns et al., 2018*).

In addition to ubiquitin ligases, pathogens also employ DUBs to hijack host cellular processes for their benefits (*Wilkinson, 2009*) and a number of DUBs have been found in *L. pneumophila* (*Kitao et al., 2020*). Among these, members of the SidE effector family harbor DUB activity in a domain that encompasses their first 200 residues, which appears to exhibit preference for K63-linked ubiquitin chains (*Sheedlo et al., 2015*). LotA is a DUB with two distinct catalytic domains, one of which functions to cleave K6-linked polyubiquitins specifically and the other displaying general DUB activity toward polyubiquitins regardless of their structure (*Kubori et al., 2018*). More recently, we have identified Ceg23 as a DUB that exclusively cleaves K63-linked polyubiquitin chains (*Ma et al., 2020*). This emerging list of DUBs likely reflect an expansive interplay of *L. pneumophila* with the host ubiquitin system mediated by its extensive suite of ubiquitin ligases; there are likely more DUBs in its effector repertoire for a controlled interference with the host ubiquitin system. Driven by the hypothesis that additional DUBs may contribute to the modulation of the ubiquitin network by *L. pneumophila*, we performed bioinformatic analyses of Dot/Icm substrates (*Burstein et al., 2009*; *Zhu et al., 2011*) to identify proteins of potential DUB activity. This exercise led to the identification of Lem27 (Lpg2529) which harbors a motif distantly similar to members of the OTU superfamily of DUBs. Our results demonstrate that Lem27 displays a broader activity toward ubiquitin chain types with a distinct preference for K6-, K11- and K48-type polyubiquitin chains. Structural analysis reveals that Lem27 harbors an OTU1 fold and that it recognizes ubiquitin via interactions mediated by at

least six contact sites. We also provide evidence to show that Lem27 coordinates with the SidC family E3 ligases to regulate protein ubiquitination on the surface of bacterial phagosomes.

## Results

### Identification of Lem27 as a DUB

To identify additional effectors that potentially harbor DUB activity, we analyzed Dot/Icm substrates (*Burstein et al., 2009*; *Zhu et al., 2011*) with HHpred (*Soding et al., 2005*) and found that Lem27 contains a motif remotely resembling active sites associated with the OTU superfamily of DUBs (*Balakirev et al., 2003*). Sequence alignment produced by the HHpred algorithm suggests that Cys24 and two adjacent residues of Lem27 constitutes a conserved Gly-Asn-Cys tripeptide motif shared among several cysteine proteases involved in the regulation of modifications by ubiquitin or the ubiquitin-like ISG15 from humans or viral pathogens (*Figure 1*).

To probe the DUB activity of Lem27, we first examined its reactivity with ubiquitin-propargylamide (Ub-PA), a commonly used substrate-based inhibitor that forms a covalent linkage with the active site Cys residue of various DUBs (*Galardy et al., 2005*), including members of the UCH, USP and OTU families (*Figure 2a* upper panel) (*Mevissen et al., 2013*). Incubation of Lem27 with Ub-PA led to the formation of a covalent conjugate typical for reactive DUBs, which is characterized by an approximately 8 kDa increase in its molecular weight (*Figure 2a*, lower panel). We further examined the reactivity by creating Lem27$_{C24A}$, in which the predicted active cysteine was substituted with alanine. This mutant has lost the ability to react with Ub-PA (*Figure 2a*, lower panel). These results suggest that Lem27 is a DUB which uses Cys24 for catalysis.

To further examine the DUB activity of Lem27, we cotransfected HEK293T cells with plasmids that direct the expression of Flag-Ub and GFP-Lem27, respectively, and proteins ubiquitinated by Flag-Ub were probed by immunoblotting with the Flag-specific antibody. SdeA$_{DUB}$, the amino terminal portion of SdeA with canonical DUB activity (*Qiu and Luo, 2017b*; *Sheedlo et al., 2015*) was used as a control. In cells coexpressing GFP alone, a robust modification of cellular proteins via ubiquitination by Flag-Ub was detected (*Figure 2b*). In contrast, although Flag-Ub was expressed at comparable levels, ubiquitinated proteins in cells coexpressing GFP-Lem27 or GFP-SdeA$_{DUB}$ were

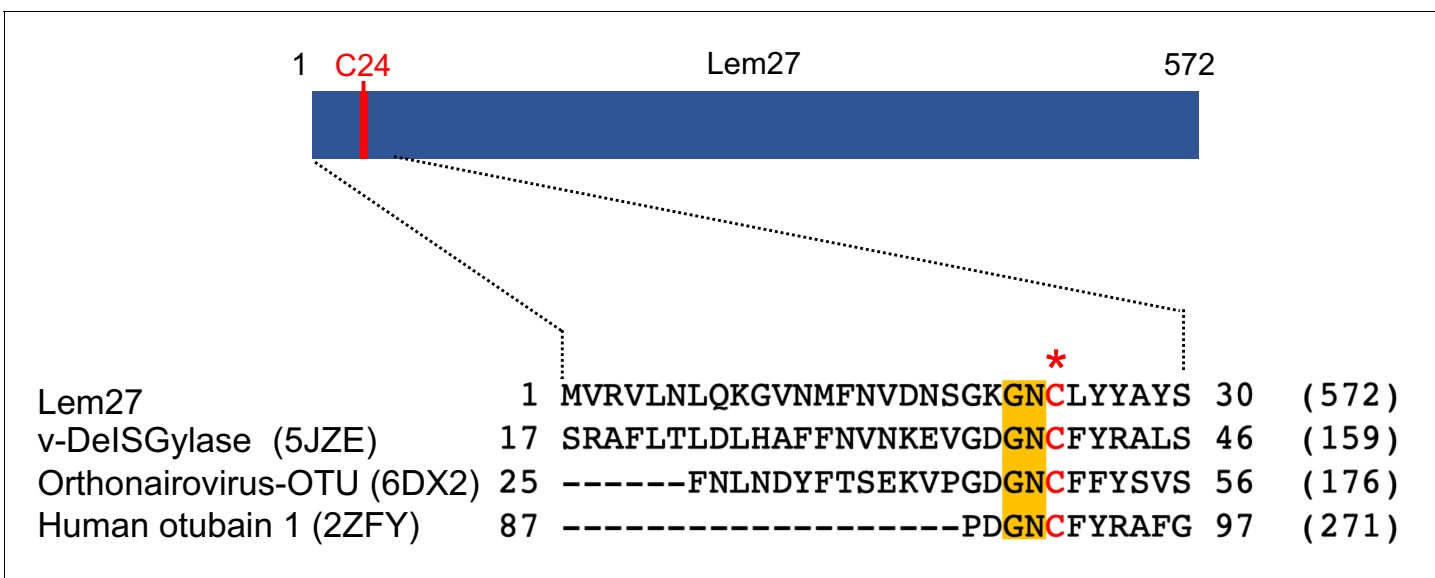

**Figure 1.** Sequence alignment of Lem27 with proteases involved in removing ubiquitin or ISG15 from modified proteins. The protein sequence of Lem27 was used as a search query for HHpred (https://toolkit.tuebingen.mpg.de/tools/hhpred) analysis. Proteins retrieved by the search known to have protease activity relevant to ubiquitin were aligned manually. The PDB codes for these proteins are in the parentheses. The cysteine residue critical for catalysis in red letter was indicated by an asterisk. Conserved residues adjacent to the catalytic cysteine were highlighted by an orange background. The numbers at the ends of the sequences indicate the positions of the residues in the proteins and the numbers in the parentheses are the lengths of the proteins.

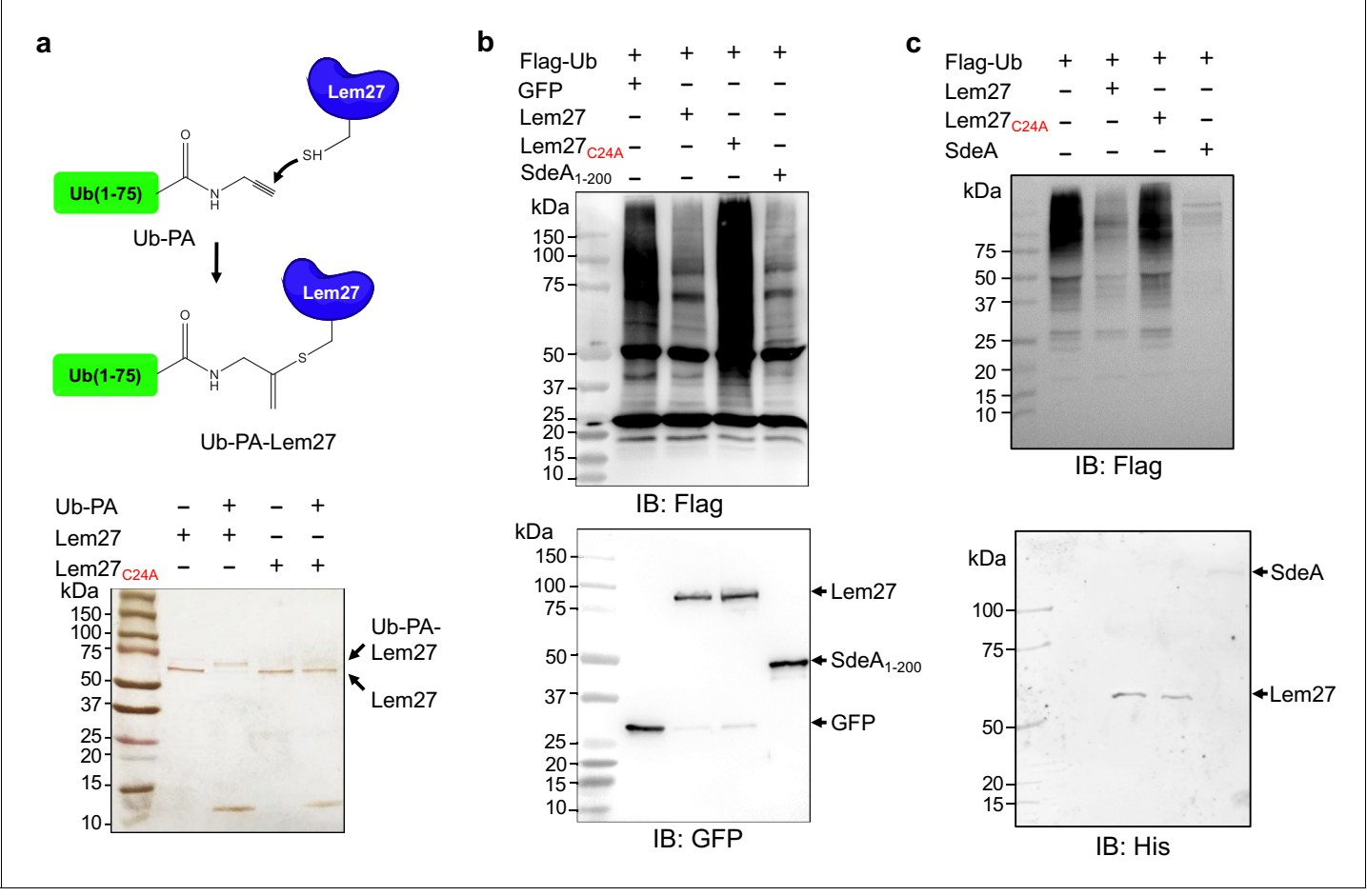

**Figure 2.** Lem27 is a deubiquitinase whose activity requires Cys24. (a) The formation of a covalent conjugate between Lem27 and the DUB inhibitor Ub-PA. A diagram showing the chemical reaction between the reactive end of Ub-PA and the side chain of Cys24 from Lem27 (upper panel). Lem27 or Lem27$_{C24A}$ was incubated with Ub-PA and the products resolved by SDS-PAGE were detected by silver staining (lower panel). Note the molecular weight shift of Lem27 after reacting with Ub-PA and the inability of Lem27$_{C24A}$ to cause such shift. (b) Lem27 interferes with protein ubiquitination in cells. HEK293T cells were transfected to coexpress Flag-Ub and GFP-Lem27, GFP-Lem27$_{C24A}$ , or GFP-SdeA$_{1-200}$. Proteins modified by Flag-Ub were detected by immunoblotting with a Flag-specific antibody (upper panel). The expression of the DUBs and their mutants were detected with GFP antibodies by immunoblotting. (c) Recombinant Lem27 removes ubiquitin from modified proteins. Ubiquitinated proteins isolated by immunoprecipitation from cells transfected to express Flag-Ub were incubated with His$_6$-Lem27, His$_6$-Lem27$_{C24A}$, or His$_6$-SdeA. Ubiquitination signals were detected by immunoblotting with a Flag-specific antibody (upper panel); recombinant proteins used in the reactions were detected with a His$_6$-specific antibody (lower panel).

considerably reduced (*Figure 2b*). In agreement with the DUB activity observed in the cotransfection experiments, purified His$_6$-Lem27 but not its inactive mutant His$_6$-Lem27$_{C24A}$ effectively removed ubiquitin from proteins ubiquitinated by Flag-Ub that were isolated from cells by immunoprecipitation (*Figure 2c*). Together, these results indicate that Lem27 is a DUB whose activity requires Cys$_{24}$.

## Lem27 impacts the association of ubiquitinated proteins with the LCV

The LCV is enriched with ubiquitinated proteins (*Dorer et al., 2006*), which likely is dynamically regulated by enzymes involved in ubiquitination from both the host and the pathogen. To explore the potential role of Lem27 in this process, we first determined the cellular localization of Lem27 in macrophages infected with *L. pneumophila*. To this end, we first constructed a plasmid to express 4xFlag-Lem27 in relevant *L. pneumophila* strains. Next, we determined the translocation of 4xFlag-Lem27 into host cells by fractionation using U937 cells infected with the bacterial strains after lysis by saponin, a detergent that damages the membranes of mammalian but not bacterial cells (*VanRheenen et al., 2006*). Flag-Lem27 was detected in the saponin-soluble fraction of U937 cells

infected with a strain harboring a functional Dot/Icm system but not a strain lacking an active Dot/Icm transporter (*Figure 3a*), validating that Lem27 is translocated into host cells by the Dot/Icm transporter during *L. pneumophila* infection.

To determine the cellular localization of Lem27 in infected cells, we first differentially labeled extracellular and phagocytosed bacteria with antibodies specific for *L. pneumophila*, followed by immunostaining with the Flag-specific antibody to detect translocated Lem27. Clear staining signals detected by the Flag antibody were concentrated on the LCV (*Figure 3b–c*). Such association only occurred in cells infected with the *L. pneumophila* strain harboring a functional Dot/Icm system, with approximately 80% staining positive for the fusion protein (*Figure 3b–c*). Although 4xFlag-Lem27

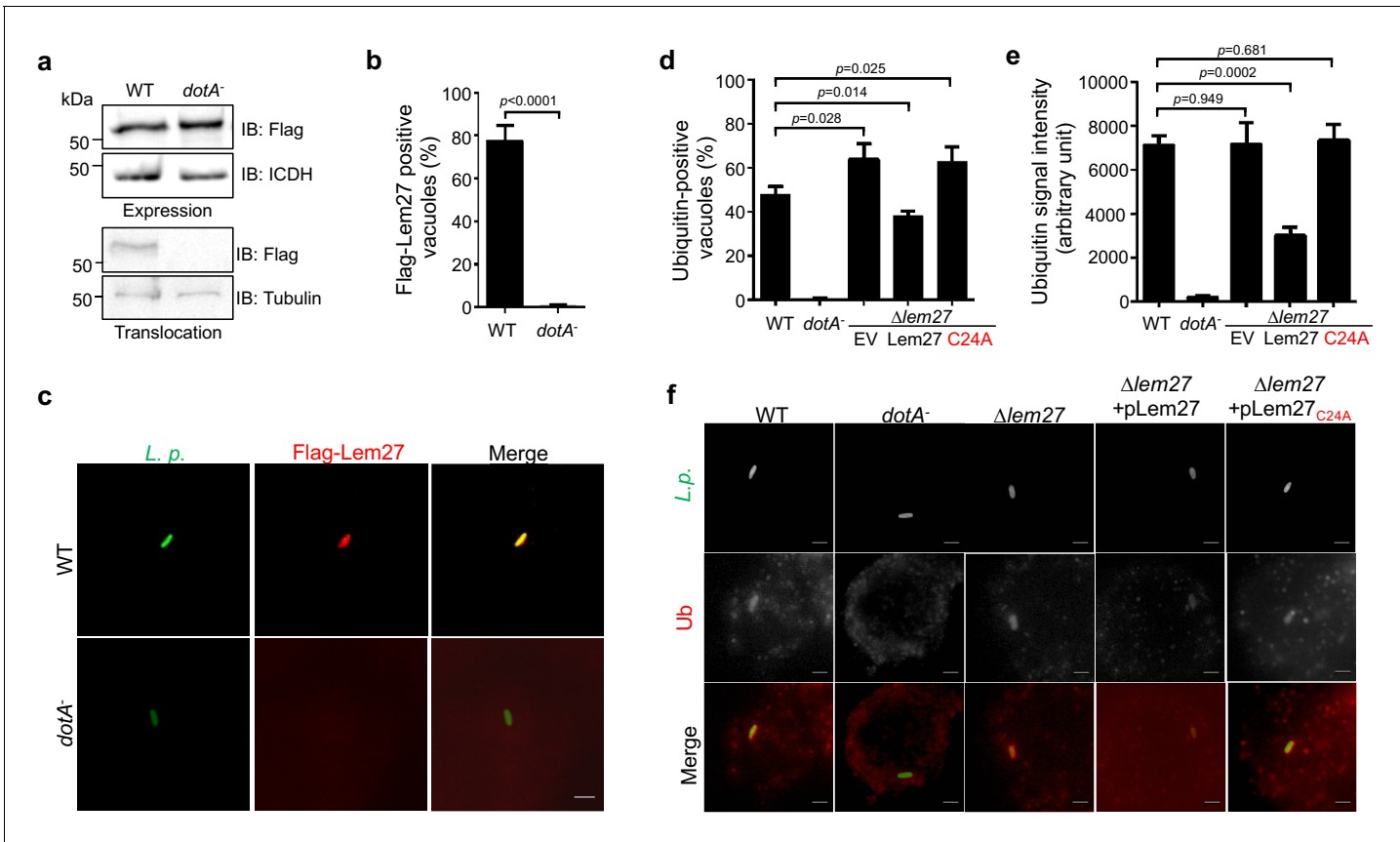

**Figure 3.** Lem27 impacts the association of ubiquitinated proteins on the LCV. (a) Lem27 is injected into host cells during *L. pneumophila* infection. U937 cells infected with bacterial strains expressing 4xFlag-Lem27 were lysed with saponin and detected for the proteins of interest in soluble and insoluble fractions, respectively. Note that the protein was expressed comparably in these two strains but it was detected only in cells infected with the strain harboring a functional Dot/Icm system. (b–c) Quantitation of the association of 4xFlag-Lem27 with the LCV and representative images of ubiquitin decorated phagosomes. U937 macrophages infected with the indicated bacterial strains for 2 hr were subjected to immunostaining and the number of vacuoles stained positive by the Flag antibody was determined (b). At least 150 vacuoles were scored for each sample and similar results were obtained in three independent experiments. Data shown were mean ±s.e. Images (c) were acquired with an Olympus IX-83 fluorescence microscope. Bar: 2 µm. (d–f) Lem27 regulates the association of ubiquitinated proteins on the LCV. Macrophages infected with the indicated *L. pneumophila* strains were immunostained to identify the bacterial vacuoles followed by staining with the FK1 ubiquitin antibody. The percentage of ubiquitin positive vacuoles was scored by counting at least 150 intracellular bacterial (d) and the intensity of the ubiquitin staining signal of the scored vacuoles was measured (e). Results shown were mean ± s.e. from three independent experiments. Representative images of the association of ubiquitin with LCVs containing relevant bacterial strains (f). Images were acquired with an Olympus IX-83 fluorescence microscope. Bar: 2 µm.

The online version of this article includes the following figure supplement(s) for figure 3:

**Figure supplement 1.** The association of Lem27 with *L. pneumophila* phagosomes in *D. discoideum D. discoideum* cells seeded on coverslips were infected with wild-type *L. pneumophila* or the *dotA* mutant expressing.

**Figure supplement 2.** Expression of *lem27* in *L. pneumophila* grown at different phases.

**Figure supplement 3.** Intracellular growth of *L. pneumophila* mutants lacking one or more bacterial deubiquitinase genes in host cells.

was similarly expressed in the *dotA*⁻ strain, association of the Lem27 with its vacuoles was not detected (*Figure 3b–c*).

We also infected the protozoan host *D. discoideum* with these *L. pneumophila* strains and examined the association of Lem27 with the LCV. For the strain expressing a functional Dot/Icm system, approximately 80% of the vacuoles stained positive for Lem27 (*Figure 3—figure supplement 1*) and no such association was detected in samples infected with the strain lacking a functional transporter (*Figure 3—figure supplement 1*).

The localization of Lem27 on the bacterial phagosome prompted us to examine whether it plays a role in the association of ubiquitinated proteins with the LCV. U937 macrophages infected with relevant *L. pneumophila* strains for 2 hr were subjected to immunostaining with the FK1 antibody specific for ubiquitinated proteins (*Kubori et al., 2018*). As expected, close to 40% of the vacuoles containing the wild-type strain Lp02 stained positive by the antibody and no staining signal was detected in vacuoles containing the *dotA*⁻ mutant Lp03 (*Figure 3d–e*). Importantly, we observed a significant increase in the percentage of vacuoles positive for ubiquitin in infections with the strain lacking the *lem27* gene (*Figure 3d and f*). Furthermore, expression of *lem27* but not the catalytically inactive mutant *lem27*$_{C24A}$ in the mutant strain from a plasmid restored the percentage of association to wild-type levels (*Figure 3d and f*). Similar results were obtained when the intensity of the ubiquitin staining signal was examined (*Figure 3e*). These results indicate that Lem27 functions to regulate protein ubiquitination on the LCV.

## Expression of Lem27 is induced at the transmissive phase, and this gene is dispensable for bacterial intracellular replication in macrophages

To accommodate the need of effector activity at different phases of its interactions with host cells, *L. pneumophila* temporally regulates the expression of many of its effectors in response to various signals (*Segal, 2013*). For example, a large number of effector genes are induced at late exponential phase (*Isberg et al., 2009*), which may allow effective subversion of host processes when the pathogen makes the initial contact with the host cell. We used Lem27-specific antibodies to examine the expression pattern of *lem27* by monitoring its protein levels in bacterial cells grown at different phases in bacteriological media. Lem27 was readily detectable in freshly diluted bacteria, which was maintained at similar levels into the exponential phase. The protein level began to increase when the culture entered later exponential phase (12 hr) and peaked at the stationary phase (*Figure 3—figure supplement 2*). Thus, similar to many Dot/Icm substrates (*Nagai et al., 2002*; *Luo and Isberg, 2004*; *VanRheenen et al., 2006*), *lem27* is induced in bacteria of the transmissive state, which may contribute to overcome host defense in the initial phase of infection. In addition, given its relatively high-level expression at other growth phases, including the exponential phase (*Figure 3—figure supplement 2*), Lem27 likely also plays a role in other stages of the intracellular life cycle of *L. pneumophila*.

To determine the role of *lem27* in intracellular bacterial replication, we infected RAW264.7 macrophages or the protozoan host *Dictyostelium discoideum* with the Δ*lem27* mutant and relevant *L. pneumophila* strains. The mutant grew indistinguishably to that of the wild-type strain in both hosts (*Figure 3—figure supplement 3a–c*), indicating that similarly to the majority of characterized Dot/Icm substrates, deletion of *lem27* did not detectably affect intracellular replication of the bacterium in laboratory infection models.

A few DUBs have been described in *L. pneumophila* (*Kitao et al., 2020*) of which LotA (*Kubori et al., 2018*) and Ceg23 (*Ma et al., 2020*) also belong to the OTU superfamily. Furthermore, LotA exhibits preference toward several chain types, including K6-type (*Kubori et al., 2018*), which is one of the preferred chain types cleaved by Lem27 (see below). We thus examined the potential functional redundancy between *lotA* and *lem27* by constructing a mutant lacking both genes. This strain Lp02Δ*lotA*Δ*lem27*, did not display detectable growth defects in either mammalian macrophages or *D. discoideum* (*Figure 3—figure supplement 3d*).

## Lem27 preferentially cleaves diubiquitin linked by K6, K11, or K48

Polyubiquitin chains of distinct architecture are formed by isopeptide bonds established by one of the seven lysine residues, K6, K11, K27, K29, K33, K48, K63 as well as the amino group of the

N-terminal methionine (M1) of a preceding ubiquitin and the carboxylate of Gly76 of the succeeding monomer, resulting in eight types of homotypic polyubiquitin chains (*Komander and Rape, 2012*). To determine whether Lem27 prefers certain specific chain types, we tested its ability to cleave the eight different types of diubiquitin. In reactions containing 1.5 μM substrate and 1.0 μM His$_6$-Lem27, approximately 50% of the diubiquitin linked at K6, K11 or K48-linked diubiquitin substrates was efficiently cleaved in 10 min. In the same reaction time, approximately 35% of diubiquitins linked at K63 or K33 in the reactions was cleaved and cleavage of diubiquitin linked at K27 or K29 was barely detectable (*Figure 4a*). When the reactions were allowed to proceed for 2 hr, cleavage of K6-linked diubiquitin was close to 80%, those linked at K63 or K33 were cleaved to about 65% and the cleave of K27-linked diubiquitin was around 15%. Extension of the reaction time to 2 hr only led to negligible cleavage of K29-linked diubiquitin (*Figure 4a*). Similarly, cleavage of M1-linked linear diubiquitin could not be detected even in reactions that were allowed to proceed for 2 hr (*Figure 4b*). As expected, the catalytically inactive mutant Lem27$_{C24A}$ has lost the ability to cleave any of these substrates (*Figure 4a–b*). Thus, Lem27 exhibits certain degree of selectivity toward its substrates, with a preference for polyubiquitin chains linked at K6, K11, or K48, followed by K33- and K63- types. In contrast, diubiquitin linked at K29 and linear diubiquitin cannot be cleaved by Lem27 (*Figure 4a–b*).

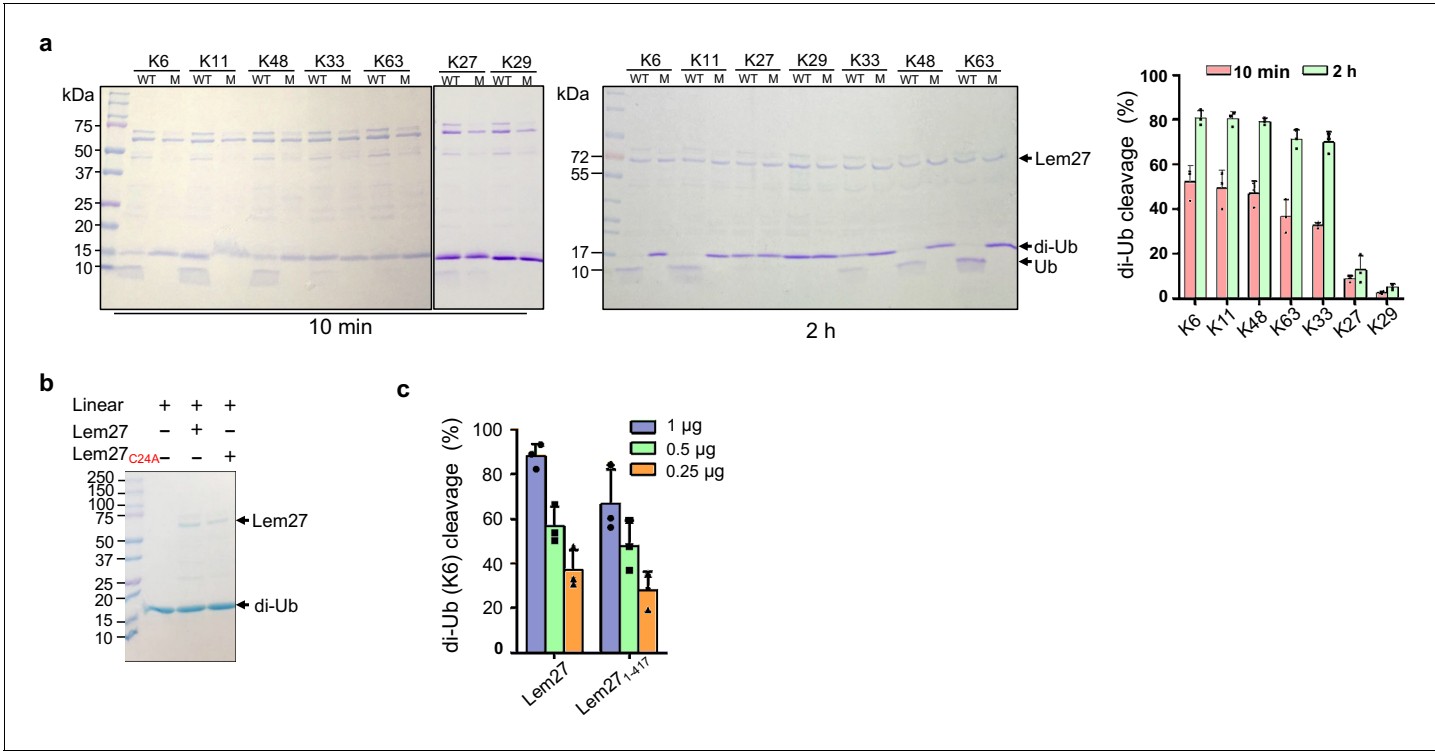

**Figure 4.** Substrate preference of Lem27. (a) Diubiquitin linked by K6, K11 or K48 was the preferred substrates for Lem27. A representative image of diubiquitin digestion by Lem27 in which recombinant Lem27 or Lem27$_{C24A}$ was incubated with the indicated diubiquitin for 10 min (left panel) or 2 hr (middle panel). The cleavage of the substrates was detected by Coomassie brilliant blue (CBB) staining after SDS-PAGE. The percentage of cleavage for each tested diubiquitin was calculated from three independent experiments (right panel). (b) Lem27 cannot cleave linear diubiquitin. Recombinant Lem27 or Lem27$_{C24A}$ was incubated with linear diubiquitin for 2 hr and proteins in the reactions were separated by SDS-PAGE and detected by CBB staining. (c) A Lem27 deletion mutant lacking 155 residues from its carboxyl end is active. Reactions containing K6-linked diubiquitin and the indicated amounts of Lem27 or Lem27$_{1-417}$ was allowed to proceed for 10 min prior to SDS-PAGE and CBB staining. The percentage of cleavage shown were from three independent experiments. Data shown were mean ±s.e.

The online version of this article includes the following figure supplement(s) for figure 4:

**Figure supplement 1.** The overall structure of the Lem27$_{1-417}$-ubiquitin complex in one asymmetric unit.

# Lem27 recognizes ubiquitin by six contact sites via multiple hydrogen bonds

To understand the molecular details of Ub recognition and catalysis by Lem27, we aimed to determine its crystal structure. We found that the full-length Lem27 construct has the propensity to form aggregates of different sizes in solution, we therefore examined several carboxyl-terminal truncation mutants of which the $Lem27_{1-417}$ construct remained active against diubiquitin (*Figure 4c*). Furthermore, this truncation mutant produced homogenous samples in its covalent conjugate form with Ub-PA. Upon subjecting the complex formed by selenomethionine-labeled $Lem27_{1-417}$ and Ub-PA to crystallization screening, we obtained crystals and solved the structure to a resolution of 2.43 Å by single-wavelength anomalous diffraction (SAD) phasing (*Table 1*).

There are two copies of the $Lem27_{1-417}$-Ub-PA complex in an asymmetric unit (ASU)(*Figure 4—figure supplement 1a*). However, structural analysis suggests that there is no intermolecular interaction between these two $Lem27_{1-417}$ molecules, which is consistent with the results from PISA analysis (average interface area of 811 $Å^2$ between the two $Lem27_{1-147}$ subunits in the ASU vs 1904 $Å^2$ of average interface area between Ub-PA and $Lem27_{1-417}$ within one complex). The two DUB modules of $Lem27_{1-417}$ molecules in an ASU are almost identical with a root-mean-square deviation (RMSD) of 0.249 Å over Cα atoms of 407 residues (residues 1–417), except that there is an approximate 35° rotation of a domain consisting of a bundle of four α-helices (α-bundles) formed by α13-α16 spanning residues 315–417 (*Figure 4—figure supplement 1b*). However, these α-bundles collectively behave as a rigid body with an RMSD about 0.51 Å over Cα of 97 residues, implying that the linker between the α-bundles and the DUB domain is flexible (*Figure 4—figure supplement 1c*). The $Lem27_{1-417}$ molecule is composed of two distinct domains (*Figure 5a*): a DUB domain spanning the first ~300 amino acids (residues 1–314) and a separate domain consisting of the four α-bundles (residues 315–417) (*Figure 5b*). The α-bundle, while sharing limited contacts with the DUB domain,

**Table 1.** Data collection and refinement statistics.

| Data collection | SeMet Lem27$_{(1-417)}$-Ub-PA |
| --- | --- |
| Wavelength (Å) | 0.9792 |
| Space group | *P 1 2$_1$ 1* |
| Cell dimensions | |
| a, b, c (Å) | 66.75,118.95, 84.28 |
| α, β, γ (°) | 90.00, 98.28, 90.00 |
| Resolution (Å) | 31.95–2.43 (2.52–2.43) |
| No. of reflections | 49000 (4827) |
| R$_{merge}$ | 0.116 (0.661) |
| R$_{pim}$ | 0.049 (0.324) |
| I/σI | 9.20 (2.10) |
| Completeness (%) | 99.83 (99.42) |
| Redundancy | 6.6 (5.10) |
| Refinement | |
| Resolution (Å) | 31.950–2.43 (2.52–2.43) |
| No. reflections | 48979 (4824) |
| R$_{work}$/R$_{free}$ (%) | 22.27 (33.48)/22.69 (33.46) |
| Total no. of atoms | 7947 |
| Ramachandran plot | |
| Wilson B-factor (Å) | 49.28 |
| Favoured (%) | 96.03 |
| Allowed (%) | 3.77 |
| Outliers (%) | 0.21 |

Values in parentheses are for the highest resolution shell.

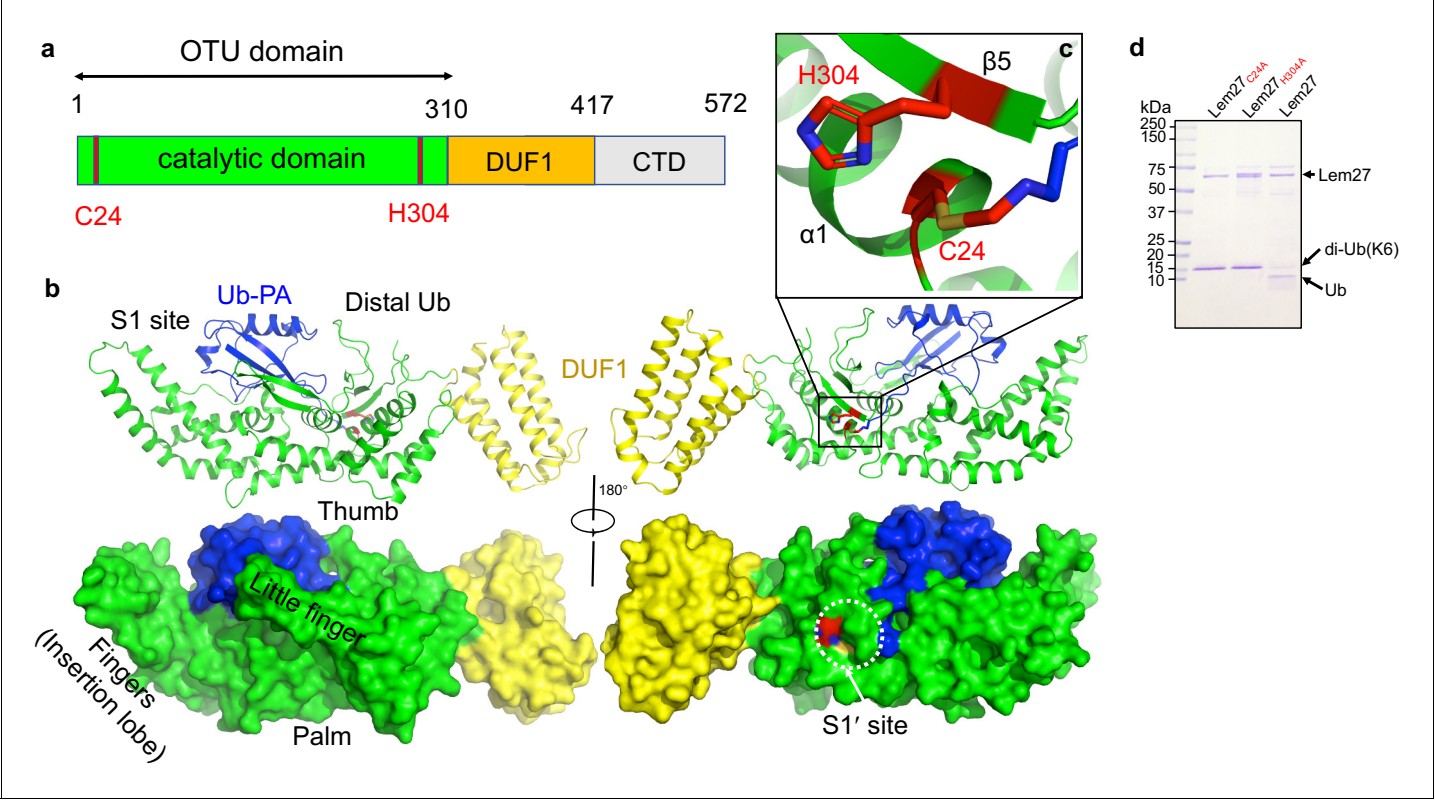

**Figure 5.** The structure of the covalent complex formed by Lem27$_{1-417}$ and Ub-PA. (a) Domain organization of full-length Lem27, the DUB module (residues 1–310), the DUF1 (residues 315–417) and the carboxyl terminus domain (CTD, not available in the structure) are shown. (b) Two different views of the overall structure of Lem27$_{1-417}$-Ub-PA. The ribbon mode of Lem27$_{1-417}$-Ub-PA binary complex (upper panel) and surface model (lower panel). Distinct portions of the hand-shaped structure and the S1 and S1' sites were indicated. (c) The configuration of the catalytic center formed by Cys24 and His304 of Lem27. The side chain of Cys24 has formed a chemical bond with Ub-PA. (d) Residues Cys24 and His304 are essential for the enzymatic activity of Lem27. His$_6$-Lem27, His$_6$-Lem27$_{C24A}$, or His$_6$-Lem27$_{H304A}$ was incubated with K6-type diubiquitin and the protein products were detected by CBB staining after SDS-PAGE. Similar results were obtained in at least three independent experiments.

The online version of this article includes the following figure supplement(s) for figure 5:

**Figure supplement 1.** Structural comparison between Lem27 and several DUBs.

appears as an appendage to the DUB domain and may serve as an independent unit, whose function is unclear and was therefore designated as domain of unknown function 1 (DUF1). The DUB module with a bilobal architecture assumes the shape of the right hand with one of the lobes giving the appearance of the thumb and forefinger and the other lobe as palm plus the three other fingers (*Figure 5b*). One lobe is composed of six α-helices (1–4, 10, and 11) and five β-strands (1-5) where the strands form two separate β-sheets: one comprising of strands 1–3 and the other is an antiparallel β-hairpin (strands 4 and 5) which protrudes out from the rest of the mixed α-β core in this lobe (*Figure 5b*). The second lobe is composed of five α-helices α5 to α9, which form the fingers part of the hand (finger lobe). The protruding hairpin appears as the thumb with the rest of the structure as the palm. The ubiquitin moiety is being held tightly by the open hand with its carboxyl end being inserted into the catalytic cleft to access the side chain of Cys24. A vinylthioether bond is installed between the side chain of Cys24 from Lem27$_{1-417}$ and the propargylamide reactive end of Ub-PA (*Figure 5c*). In the structure, His304 is found within 3.8 Å of the S-atom of the catalytic Cys which most likely would constitute the key Cys-His catalytic dyad for Lem27 (*Figure 5c*). The proximity of His304 and its putative role in the formation of the catalytic center suggested that this residue is critical for the DUB activity of Lem27. Indeed, substitution of His304 with alanine abolishes its enzymatic activity (*Figure 5d*).

DALI search against the PDB showed that the overall structure of Lem27$_{1-417}$ possesses a folding (*Figure 5—figure supplement 1*) that shares some similarity mainly with the core folding unit of

Group I OTU DUBs found in OTUB1, OTUD3, OTUD5, and OTULIN (Table S1). Although the sequences of these enzymes vary greatly with only 13% identity at the most, the structures of the cores are similar. Furthermore, in each case positioning of the two catalytic residues (Cys-His) in these structures is highly similar (*Figure 5—figure supplement 1a*). Lem27$_{1-417}$ displays the highest similarity to OTUB1 (PDB: 4DDG) of humans among the structural homologs, with a Dali Z score of 11.0 and an RMSD of 2.32 Å over a region encompassing 143 amino acids in the core part of the structure (*Figure 5—figure supplement 1b*). Interestingly, when compared with these OTU-domain containing structures, the finger lobe of Lem27$_{1-417}$ (residues 108–221) appears as an insertion domain (residues 108–221), absent in structures of OTU DUBs of eukaryotic origin. Such an insertion domain, although considerably shorter, is present in Ceg23 (6KS5, residues 123-201aa), a K63-type specific OTU DUB from *L. pneumophila* identified in our earlier study (*Figure 5—figure supplement 1c*; *Ma et al., 2020*). Of note, the insertion domains between Lem27 and Ceg23 lack any similarity in their structures, which may account for their differences in substrate specificity.

Inspection of the Lem27$_{1-417}$-Ub-PA complex reveals that the DUB domain makes extensive contacts with the distal ubiquitin moiety in the S1 site of Lem27 by multiple pairs of hydrogen bonds with a total buried conserved surface area of approximately 1900 Å$^2$ (*Figure 5b*). A total of six distinct interaction sites can be identified in the interface between Lem27$_{1-417}$ and Ub (*Figure 6*). PDBsum analysis (*Laskowski et al., 2018*) reveals that the interactions include 30 hydrogen bonds, five salt bridges and 288 Van der Waals forces (*Figure 6—figure supplement 1a*). Site-1 is the catalytic pocket which recognizes the carboxyl-terminal tail of ubiquitin via several pairs of hydrogen bonds, including those formed between Lem27$_{E244}$ and Ub$_{L71}$, Lem27$_{S116,D148}$ and Ub$_{R72}$,

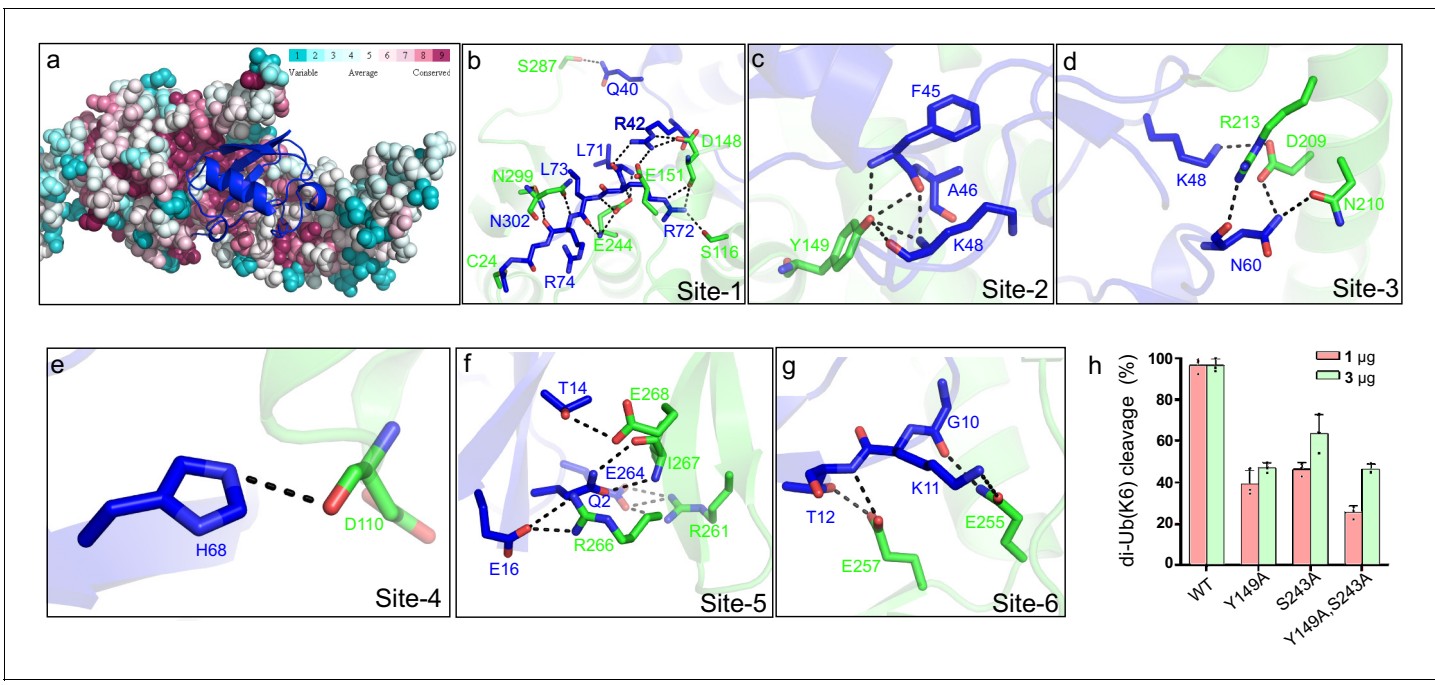

**Figure 6.** Molecular recognition of ubiquitin by the DUB module of Lem27. (**a**) Ubiquitin shown in ribbon mode binds to Lem27 DUB (shown in surface mode) via the vinylthioether bond formed by Cys24 and Ub-PA. Analysis by the ConSurf Server (http://consurftest.tau.ac.il/) shows that the ubiquitin binding sites on the palm of hand-shaped Lem27 are highly conserved. (**b**) The carboxyl terminus of ubiquitin inserts into the catalytic cleft of Lem27 DUB, which is mainly stabilized by backbone interactions, Lem27 residues involved in interaction with ubiquitin are shown as sticks. (**c**) Tyr149 of Lem27 specifically binds Phe45 and the Lys48 backbone of ubiquitin. (**d**) A hydrogen bond net is formed by Lys148, Asn60 of ubiquitin and Arg213, Asp209 and Asn210 of Lem27. (**e**) The hydrogen formed by His68 of ubiquitin and Asp110 of ubiquitin. (**f**) The interaction between the β-hairpin of Lem27 and ubiquitin, Arg266, Ile267, Glu268 of β1 of Lem27 and Arg261 of β2 are shown as sticks, the residues in ubiquitin are also shown as sticks. (**g**) The interaction between the β core of ubiquitin and Lem27. (**h**) Several residues involved in binding ubiquitin are important for substrate cleavage by Lem27. Data shown were mean ± s.e. as the percentage of cleavage of K6 diubiquitin by Lem27$_{1-147}$ from three independent experiments.

The online version of this article includes the following figure supplement(s) for figure 6:

**Figure supplement 1.** Lem27 makes extensive contacts with ubiquitin by numerous amino acids.

Lem27$_{E151,D148}$ and Ub$_{R42}$, Lem27$_{E244}$ and Ub$_{L73}$, Lem27$_{N302,N299}$ and Ub$_{R74}$, Lem27$_{V303}$ and Ub$_{G75}$ (*Figure 6b* and *Figure 6—figure supplement 1a*). These interactions appear to stabilize the carboxyl tail of ubiquitin in an extended conformation so that the scissile peptide bond is placed next to the catalytic Cys. Site-2 is centered by Lem27$_{Y149}$, which forms multiple hydrogen bonds with side chains of residues in ubiquitin, including Ub$_{F45}$ and Ub$_{K48}$. In addition, intramolecular hydrogen bonds are formed between the side chain of Ser115 of Lem27 and those of Phe45 and Ala46 (*Figure 6c*). Tyr149 is engaged in stacking interaction with the Ile44 of ubiquitin. The interactions in this patch are equivalent to the widely observed recognition of eukaryotic DUBs of the Ile44-patch of ubiquitin (*Reyes-Turcu et al., 2009*). Site-3 features hydrogen bonds between Lem27$_{D209}$ and Ub$_{K48}$, and those among Lem27$_{D209,R213,N210}$ and Ub$_{N60}$ (*Figure 6d*). Site-4 includes a pair of hydrogen bonds between Lem27$_{D110}$ and Ub$_{H68}$ (*Figure 6e*). Site-5 involves in hydrogen bonding between Lem27$_{R266,I267}$ and Ub$_{Q2}$, Lem27$_{E268}$ and Ub$_{T14}$, Lem27$_{R266}$ and Ub$_{E16}$, Lem27$_{R261}$ and Ub$_{E64}$ (*Figure 6f*). Site-6 is mediated by hydrogen bonds formed among Lem27$_{Q255,I275, L257}$ and Ub$_{K11,T12}$ (*Figure 6g*).

To evaluate the importance of these contact sites in substrate recognition and subsequent isopeptide bond cleavage, we created a total of 14 single substitution mutants in residues belonging to five of the six contact sites (*Figure 6—figure supplement 1b*). When diubiquitin linked at K6, one of the preferred substrates was used, only mutants N18A, D148A, Y149A, S243A E257A and I275A displayed significant reduction in activity (*Figure 6—figure supplement 1b*), and activity of other mutants remained comparable to that of wild-type protein (*Figure 6—figure supplement 1b*). Among these, when the molar ratio between substrate and enzyme was used as 1:1, mutants N18A, Y149A, S243A displayed less than 50% activity of the wild-type protein (*Figure 6h* and *Figure 6—figure supplement 1b*). As expected, simultaneous mutations in two independent sites, Tyr149 and Ser243 led to further loss of the activity (*Figure 6h*). Thus, Lem27 recognizes its substrate by recognizing multiple sites of the ubiquitin moiety and these sites function in concert to ensure proper recognition of its substrates.

## Lem27 and the SidC family of E3 ubiquitin ligases function in concert to regulate protein ubiquitination on bacterial phagosomes

*L. pneumophila* is known to interfere with the function host proteins by effectors of opposite biochemical activity to impose temporal or spatial regulation of their activity (*Qiu and Luo, 2017a*). Among the many ubiquitin E3 ligases coded for by *L. pneumophila*, SidC and SdcA anchor on the LCV by binding to PI(4)P (*Weber et al., 2006*). A recent study revealed that the small GTPase Rab10 is important for maximal *L. pneumophila* intracellular replication (*Jeng et al., 2019*). Furthermore, Rab10 is ubiquitinated and recruited to the LCV by the SidC family of E3 ligases (*Jeng et al., 2019*). Because SidC and its paralog SdcA are known to catalyze the synthesis of several polyubiquitin chains, including K11-type (*Hsu et al., 2014*), which is one of the chain types preferentially cleaved by Lem27 (*Figure 4*), we examined the potential interplay between the SidC family E3 ligases and Lem27 in the regulation of protein ubiquitination on the LCV. In biochemical reactions, SidC appeared to mostly induce monoubiquitination of Rab10 and such modification can be effectively reversed by Lem27; this DUB also removed ubiquitin added to the E3 ligase itself by self-ubiquitination (*Figure 7a*). Consistent with results from the earlier study (*Jeng et al., 2019*), SidC induced ubiquitination of Rab10 by the *L. pneumophila* strain Δ*sidC*Δ*sdcA*(pSidC), which expresses the E3 ligase from a multi-copy plasmid (*Figure 7b* upper panel). Importantly, infection with strain Δ*sidC*Δ*sdcA*(pSidC+pLem27), which coexpresses Lem27 led to a clear reduction in ubiquitinated Rab10 (*Figure 7b* upper panel, lanes 3 and 4). The amounts of SidC translocated into host cells among these samples are similar (*Figure 7b* lower panel), indicating that such reduction is caused by Lem27. Interestingly, when LotA, another DUB from *L. pneumophila* (*Kubori et al., 2018*) was coexpressed with SidC in strain Δ*sidC*Δ*sdcA*(pSidC), reduction in Rab10 ubiquitination did not occur (*Figure 7b* lower panel, lane 5). These results suggest that Lem27 but not LotA can counteract the activity of SidC during *L. pneumophila* infection. To further analyze the potential interplay between Lem27 and the SidC family E3 ligases during bacterial infection, we established a macrophage cell line that stably expresses mCherry-Rab10. Whereas the Rab1-mCherry signal was comparable in these cells (*Figure 7c–e*), we observed that approximately 60% of vacuoles containing wild-type bacteria recruited mCherry-Rab10 and such recruitment did not occur for vacuoles harboring a mutant defective for the Dot/Icm transporter or lacking *sidC* and *sdcA* (*Figure 7c–e*), which is in agreement

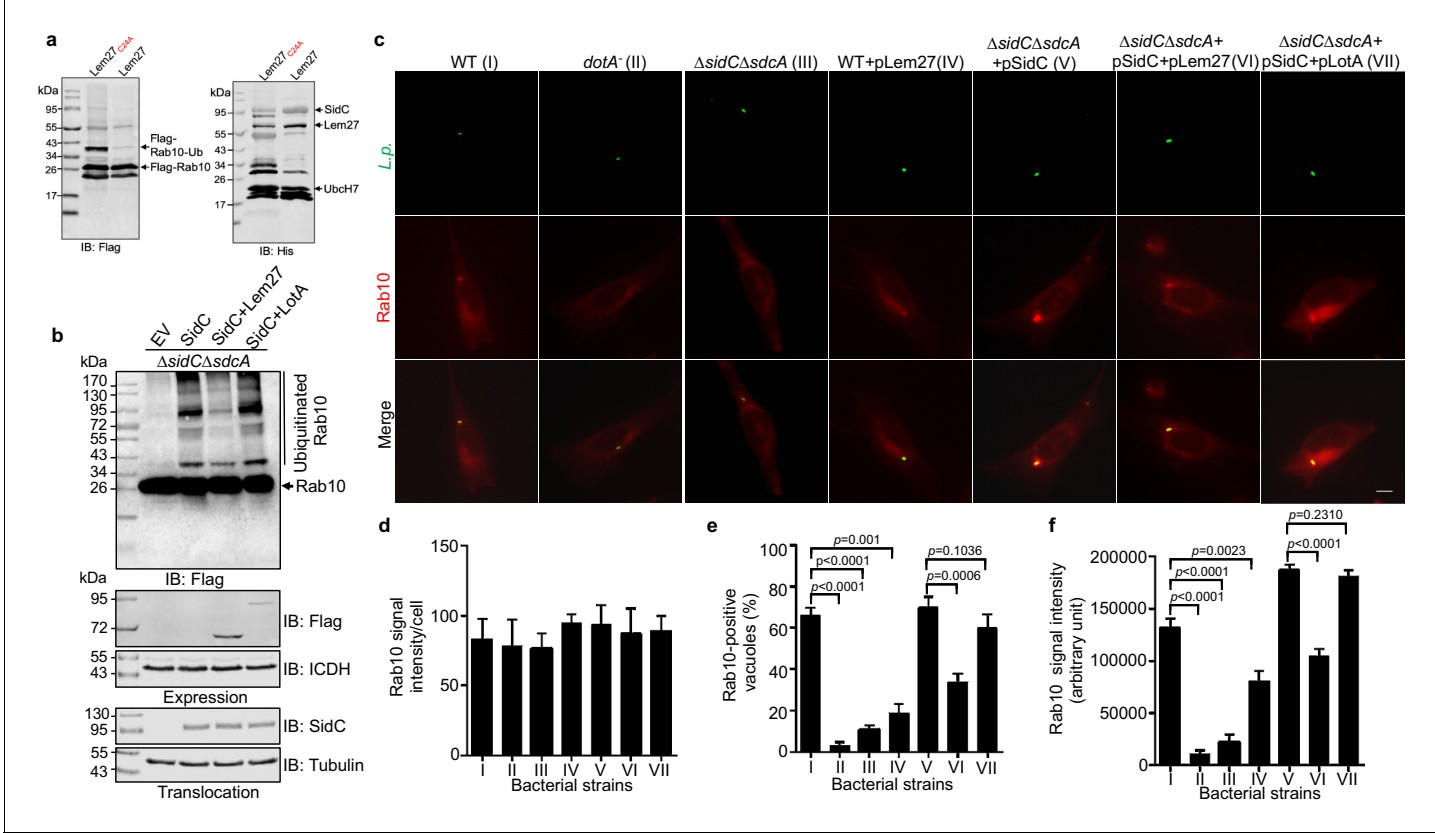

**Figure 7.** Lem27 regulates Rab10 ubiquitination and its recruitment to the LCV induced by the SidC family of E3 ubiquitin ligases. (**a**) SidC-induced Flag-Rab10 ubiquitination was reversed by Lem27. Recombinant Lem27 or its catalytically inactive mutant Lem27$_{C24A}$ was added to completed SidC-catalyzed ubiquitination reactions. Proteins resolved by SDS-PAGE were detected by immunoblotting with the Flag-specific antibody (left panel) or with His$_6$-specific antibody (right panel). Note that Lem27 removes ubiquitin from modified Rab10 and self-modified SidC (His$_6$-SidC was used for the ubiquitination reaction). (**b**) Lem27 but not LotA reduces Rab10 ubiquitination in infected cells. HEK293T cells transfected to express 4xFlag-Rab10 and the FcγII receptor for 24 hr were infected with opsonized bacteria of the indicated strains for 2 hr prior to detection by immunoblotting with Flag-specific antibody. Note that infection by the strain coexpressing Lem27 led to reduction in ubiquitinate Rab10 (3$^{rd}$ sample). The expression of Flag-Lem27 and Flag-LotA was examined in bacterial cells used for infection with the metabolic enzyme isocitrate dehydrogenase as loading controls (middle two panels) and the translocation of SidC into infected cells by each of the testing strains were detected in saponin-solubilized infected cells with tubulin as loading controls (lower two panels). (**c-f**) Lem27 but not LotA interferes with Rab10 recruitment to the LCV. A macrophage cell line stably expressing mCherry-Rab10 was infected with the indicated *L. pneumophila* strains (strain I to VII) for 2 hr and the cells were fixed for immunostaining to identify the bacterial vacuoles. Images were acquired using an Olympus IX-83 fluorescence microscope (**c**). Bar: 5 μm. The average intensity of mCherry-Rab10 signals per cell in each sample (**d**), the rates of Rab10-positive LCVs (**e**) and the intensity of Rab10 signals (**f**) on the LCVs formed by each bacterial strain were determined. For each sample at least 150 vacuoles were measured. Data shown were mean ±s.e. from three independent experiments.

with an earlier study (*Jeng et al., 2019*). Introduction of a plasmid expressing SidC into strain Δ*sidC*Δ*sdcA* not only restored the association of mCherry-Rab10 with the LCV but also significantly increased the intensity of the fluorescence signals on bacterial phagosomes (*Figure 7c,e–f*, strain V). Importantly, although not able to eliminate mCherry-Rab10 from the LCV, coexpression of Lem27 in this bacterial strain significantly reduced the percentage of Rab10-positive vacuoles and the intensity of mCherry signals on LCVs (*Figure 7e–f*, strain VI). Similar to the results of ubiquitinated Rab10 in infected cells probed by immunoblotting, co-expression of LotA with SidC did not significantly reduce the rate Rab10-positive vacuoles or the intensity of Rab10 signals on the vacuoles (*Figure 7e–f*, strain VII). Taken together, these results suggest that deubiquitination of Rab10 by Lem27 decreased its association with the LCV and that this DUB specifically regulates the activity of SidC and SdcA during *L. pneumophila* infection.

## Discussion

In this study, we identified Lem27 as an additional DUB, which adds to the complexity of the interference of the host ubiquitin network by *L. pneumophila*. Although the primary sequence of Lem27 does not share homology with established DUBs detectable by algorithms such as BLAST, analysis by predicted protein structures provided the hints for biochemical experiments that provided evidence for DUB activity (*Figures 1* and *2*). Lem27 represents a new category of DUB found in *L. pneumophila*, it appears to selectively cleave ubiquitin chains linked by K6, K33 and K48, which differs from DUBs in members of the SidE family that prefer K48 and K63-type polyubiquitin chains (*Sheedlo et al., 2015*), and Ceg23, which specifically cleaves K63-type polyubiquitin chains (*Ma et al., 2020*). Yet, to some extent, its substrate preference overlaps with LotA, another DUB from *L. pneumophila* that cleaves K6-linked polyubiquitin chains (*Kubori et al., 2018*). Both LotA and Lem27 are associated with the LCV (*Kubori et al., 2018*; *Figure 3c*), suggesting that multiple DUBs may coordinate to edit protein ubiquitination on the bacterial phagosome. Alternatively, DUBs with similar substrate preference may function at distinct compartments of infected cells. In addition to LotA, results reported herein underscore the importance of K6-linked polyubiquitin chains in cellular defense against *L. pneumophila* infection. The fact that both LotA and Lem27 are localized on the LCV suggests that K6-linked ubiquitination on the LCV is detrimental to the biogenesis and development of the vacuole.

Although Lem27 and Ceg23 (*Ma et al., 2020*) both belong to the OTU family of DUBs originally found in humans (*Wilkinson, 2009*), these proteins share little similarity in their structures except for the configuration of the catalytic center involved in isopeptide bond cleavage (*Figure 4—figure supplement 1*). Such differences may account for their distinct preference in polyubiquitin chains. Interestingly, the structure of Lem27 exhibits higher similarity to OTUB1 (PDB: 4DDG) of humans than to any other bacterial DUBs of the same family, including those from *L. pneumophila*. Given that both Lem27 and LotA, which is also an OTU family DUB, display selectivity for K6-linked ubiquitin chains, these DUBs may share some structural homology in regions involved in substrate recognition. One feature unique to Lem27 is its extensive contact with ubiquitin in the Lem27$_{1-417}$-Ub-PA complex, which is not seen in other DUBs (*Figure 5—figure supplement 1* and *Figure 6*; *Reyes-Turcu and Wilkinson, 2009*). Although such information is useful in determining how this DUB contacts the ubiquitin moiety of the Ub-PA inhibitor, elucidation of its substrate recognition mechanism and basis of selectivity will require structural study of the complexes formed by Lem27 and one or more diubiquitin as described in earlier studies (*Mevissen et al., 2013*). Our structure reveals that both Lem27$_{1-417}$ and Ceg23 contain an insertion domain not found in OTU DUBs from mammalian cells (*Figure 5—figure supplement 1*). Such domains may be involved in substrate recognition, akin to the recently described transglutaminase MvcA that catalyzes deubiquitination of the E2 conjugation enzyme UBE2N (*Gan et al., 2020*), and polyubiquitin linkage specificity by contributing to binding of additional ubiquitin groups aside from the S1 binding site observed here.

*L. pneumophila* codes for a large cohort of ubiquitin ligases that utilize distinct mechanisms to modify proteins by cooperating with host E1 and E2 enzymes, including U-box proteins (*Lomma et al., 2010*), F-box proteins (*Ensminger and Isberg, 2010*) and several that use novel mechanisms not found in eukaryotic cells (*Hsu et al., 2014*; *Lin et al., 2018*). It also codes for enzymes that catalyze ubiquitination via noncanonical mechanisms, including the E1-, E2-indepdent, NAD-powered reactions induced by members of the SidE family (*Bhogaraju et al., 2016*; *Qiu et al., 2016*) and a cross-link reaction mediated by transglutamination (*Gan et al., 2019a*). Ubiquitination catalyzed by both of these mechanisms is reversed by specific enzymes (*Gan et al., 2020*; *Shin et al., 2020*; *Wan et al., 2019*). Our observation that Lem27 and members of the SidC family regulate Rab10 ubiquitination indicates that the net outcome of protein modification by the classic mechanism is a result of the interplay between ligases and DUBs. In broth grown bacteria, the expression pattern of Lem27 and SidC is similar, both are induced at the post-exponential phase (*Luo and Isberg, 2004*; *Figure 3—figure supplement 2*), suggesting that the dynamic ubiquitination of Rab10 on the LCV occurs from the beginning of the infection. Because DUBs exhibit low substrate selectivity in biochemical reactions using purified proteins (*Sheedlo et al., 2015*), a DUB likely participates in the reversal of ubiquitination installed by multiple classic E3 ligases. Yet, it is of interest to note that although LotA share some preferred substrate preference with Lem27, it does not seem to interfere with Rab10 ubiquitination induced by SidC (*Figure 7*), suggesting that these DUBs

possess considerable substrate specificity under physiological conditions. Indeed, the reversal of ubiquitination induced by noncanonical mechanisms, which in all known cases, is catalyzed by specific enzymes (*Gan et al., 2020*; *Shin et al., 2020*; *Wan et al., 2019*). Of note, the genes coding for Lem27(*lpg2529*) and the SidC family (*sidC, lpg2511; sdcA, lpg2510*) are relatively close on the *L. pneumophila* chromosome, separated by only 18 open reading frames, which is similar to the physical proximity of genes coding for enzyme pairs that function together to temporally or spatially regulate the function of host proteins by counteracting biochemical activity (*Gan et al., 2020*; *Shin et al., 2020*; *Tan et al., 2011*; *Tan and Luo, 2011*; *Wan et al., 2019*).

The identification of additional DUBs from *L. pneumophila* clearly has added to our understanding of the role ubiquitination plays in its virulence and the co-option of host function. Deletion of a single DUB gene caused detectable defects in intracellular growth in laboratory infection models (*Kubori et al., 2018*; *Ma et al., 2020*; *Sheedlo et al., 2015*), further study aiming to identify DUBs with similar substrate specificity may shed light on the potential functional redundancy of these effectors. Although some of the DUBs acts on proteins modified by E3 ligases from the pathogen, others likely will act on proteins modified by the ubiquitin machinery of host cells. Future identification of the proteins targeted by these enzymes will surely provide insights into their roles in *L. pneumophila* virulence.

# Materials and methods

**Key resources table**

| Reagent type (species) or resource | Designation | Source or reference | Identifiers | Additional information |
|---|---|---|---|---|
| Strain, strain background (*L. pneumophila* Lp02) | Lp02Δ*lem27* | This paper | N/A | A *lem27* deletion mutant of strain Lp02 |
| Bacterial strain, strain background (*L. pneumophila* Lp02) | Lp02Δ*lem27*Δ*lotA* | This paper | N/A | A *lem27* and *lotA* deletion mutant of strain Lp02 |
| Bacterial strain, strain background (*L. pneumophila* Lp02) | Lp02Δ*sidC*Δ*sdcA* | *Hsu et al., 2014* | N/A | A *sidC* and *sdcA* deletion mutant of strain Lp02 |
| Strain, strain background (*L. pneumophila*) | Lp02Δ*sidC*Δ*sdcA*(pZL199) | This paper | N/A | Strain Lp02Δ*sidC*Δ*sdcA* expressing SidC |
| Strain, strain background (*L. pneumophila*) | Lp02Δ*sidC*Δ*sdcA*(pZL199 and pLem27) | This paper | N/A | Strain Lp02Δ*sidC*Δ*sdcA* expressing SidC and Lem27 |
| Strain, strain background (*L. pneumophila*) | Lp02Δ*sidC*Δ*sdcA*(pZL199 and pLotA) | This paper | N/A | Strain Lp02Δ*sidC*Δ*sdcA* expressing SidC and LotA |
| Cell line (Human) | HEK293T | ATCC | CRL-1573 | |
| Cell line (Human) | Hela | ATCC | CCL-2 | |
| Cell line (Mouse) | RAW264.7 | ATCC | TIB:71 | |
| Cell line (Human) | U937 | ATCC | CRL-1593.2 | |
| Cell line (Mouse) | MLE | ATCC | CRL-2110 | |
| Antibody | Mouse monoclonal ANTI-FLAG antibody M2 | Sigma | Cat. #: F1804 | WB (1: 3000) IF (1: 200) |
| Antibody | Mouse monoclonal ANTI-GFP antibody | Sigma | Cat. #: SAB5300167 | WB (1:5000) |
| Antibody | Mouse monoclonal ANTI-His antibody | Sigma | Cat. #: H1029 | WB (1: 10,000) |
| Antibody | Mouse monoclonal Anti-HA antibody | Santa Cruz | Cat. #: sc-7392 | WB (1: 1000) |

*Continued on next page*

*Continued*

| Reagent type (species) or resource | Designation | Source or reference | Identifiers | Additional information |
|---|---|---|---|---|
| Antibody | Rabbit polyclonal Anti-ICDH antibody | *Xu et al., 2010* | N/A | WB (1: 20,000) |
| Antibody | Mouse monoclonal Anti-tubulin antibody | DSHB | E7 | WB (1: 10,000) |
| Antibody | Rabbit polyclonal Anti-Lem27 antibodies | This paper | N/A | WB (1:500) |
| Antibody | Rabbit polyclonal Anti-SidC antibodies | *Luo and Isberg, 2004* | N/A | WB (1:10000) |
| Antibody | Rabbit polyclonal Anti- *L. pneumophila* antibodies | *Xu et al., 2010* | N/A | IF (1:10,000) |
| Antibody | Mouse monoclonal Anti-FKI antibody | Enzo Life Science | Prod. No. BML-PW8805 | IF (1:1,000) |
| Peptide, recombinant protein | 3XFlag Peptide | Sigma-Aldrich | Cat. #: F4799 | |
| Commercial assay or kit | Quikchange kit | Agilent | Cat. #: 600670 | |
| Commercial assay or kit | TransStart Fast Pfu DNA Polymerase | TransGen, Beijing, China | Cat. #: AP221-03 | |
| Chemical compound, drug | Ubiquitin-Propargylamine (Ub-PA) | Boston Biochem | Cat. #: U-214 | |
| Chemical compound, drug | AQUApure Di-Ub Chains (K6-linked) Protein, CF | Boston Biochem | Cat. #: UC-11B-025 | |
| Chemical compound, drug | AQUApure Di-Ub Chains (K11-linked) Protein, CF | Boston Biochem | Cat. #: UC-40B-025 | |
| Chemical compound, drug | Recombinant Human Di-Ub/Ub2 WT Chains (K27-linked), CF | Boston Biochem | Cat. #: UC-61B-025 | |
| Chemical compound, drug | AQUApure Di-Ub Chains (K29-linked) Protein, CF | Boston Biochem | Cat. #: UC-81B-025 | |
| Chemical compound, drug | AQUApure Di-Ub Chains (K33-linked) Protein, CF | Boston Biochem | Cat. #: UC-101B-025 | |
| Chemical compound, drug | AQUApure Di-Ub Chains (K48-linked) Protein, CF | Boston Biochem | Cat. #: UC-200B-025 | |
| Chemical compound, drug | AQUApure Di-Ub Chains (K63-linked) Protein, CF | Boston Biochem | Cat. #: UC-300B-025 | |
| Chemical compound, drug | Antibiotic G418 | Sigma-Aldrich | Cat. #: G8168 | |
| Software, algorithm | HHpred | *Soding et al., 2005* | | |
| Software, algorithm | Adobe Photoshop CS3 Extended | Adobe | | |
| Software, algorithm | Graphpad | graphpad.com | | |
| Other | Anti-HA Magnetic Beads | MedChem Express (MCE) | Cat. #: HY-K0201 | |
| Other | Anti-flag M2 affinity gel | Sigma | Cat. #: A2220 | |

## Bacterial stains and plasmids

*Escherichia coli* strains, plasmids and primers used in this study are listed in Table S2. Derivatives of *E. coli* strains DH5α, DH5αλπ or XL1blue were used for molecular cloning. *E. coli* was grown in LB medium at 37°C. When needed, antibiotics were used at the following concentrations: Ampicillin (100 μg/mL), kanamycin (30 μg/mL), streptomycin (100 μg/mL). Unless otherwise indicated, strain BL21(DE3) was used for the production of recombinant proteins expressed from the pET series of plasmids (Novagen Sigma-Aldrich). All *L. pneumophila* strains were derived from the Philadelphia one strain Lp02 and the *dotA*⁻ mutant strain Lp03 (*Berger and Isberg, 1993*). *L. pneumophila* was cultured in liquid N-(2-acetamido)−2-aminoethanesulfonic acid (ACES) buffered yeast extract medium (AYE) or on solid charcoal buffered yeast extract medium (CYE). When necessary, thymidine was added at 0.2 g/mL. Plasmids derived from pZL507 (*Xu et al., 2010*) were maintained in *L. pneumophila* by thymidine autotrophic. Gene deletion in *L. pneumophila* was carried out as described previously (*Liu and Luo, 2007*). Restriction enzymes and T4 DNA ligase were purchased from NEB. Polymerase chain reaction (PCR) amplification was performed using TransStart Fast *Pfu* DNA polymerase (AP221-03, TransGen, Beijing, China).

Site-directed mutagenesis was performed by the Quikchange kit (Agilent) with primer pairs designed to introduce the desired mutations. The sequences of primers and plasmids made in this study are listed in Table S2. All substitution mutants were verified by double strand DNA sequencing.

## Protein expression and purification

*E. coli* strains for protein production were inoculated in LB medium containing the appropriate antibiotics and grown to saturation at 37°C in a shaker (250 rpm) overnight. The culture was diluted at 1:50 in fresh LB and isopropyl β-D-1-thiogalactopyranoside (IPTG) was added to a final concentration of 0.2 mM when $OD_{600}$ of the culture reached 0.6. The induction was allowed to proceed in a shaker (180 rpm/min) at 16°C for 16–18 hr. Bacterial cells were then harvested by centrifugation at 4000x$g$ for 15 min.

Bacterial cells were suspended in 30 mL lysis buffer (50 mM $NaH_2PO_4$, 300 mM NaCl, 10 mM imidazole, pH 8.0) and lysed using a JN-Mini Low Temperature Ultrahigh Pressure Continuous Flow Cell Cracker (JNBIO, Guangzhou, China). The soluble fraction containing the protein of interest was obtained by centrifugation at 12000x$g$ for 20 min and was mixed with $Ni^{2+}$-NTA beads (Qiagen) for 1.5 hr by rotation at 4°C. The beads were loaded onto a column and unbound proteins were removed by washing with 3 times of column volumes of washing buffer (50 mM $NaH_2PO_4$, 300 mM NaCl, 20 mM imidazole, pH 8.0). Bound $His_6$-tagged proteins were eluted with 5 mL of elution buffer (50 mM $NaH_2PO_4$, 300 mM NaCl, 250 mM imidazole, pH 8.0). The purified proteins were dialyzed in a buffer containing 25 mM Tris-HCl (pH7.5), 150 mM NaCl and 10% (v/v) glycerol. Protein concentration was determined using the Bradford assay with BSA as the standard.

To purify protein for structural determination, DNA fragments encoding full-length Lem27 or its truncation variants including $Lem27_{1-417}$ were inserted into pET28a to generate plasmids suitable for the production of $His_6$-Lem27 or fragments of different lengths. Each construct was transformed into *E. coli* BL21(DE3) and the resulting strains were used to purify $His_6$-tagged proteins. Cells suspended in a buffer (50 mM Tris-HCl, pH 8.0, 150 mM NaCl) were lysed by ultrasonication. After centrifugation at 17,000x$g$ for 30 min, $His_6$-tagged Lem27 or its truncation mutants was purified using $Ni^{2+}$-NTA columns (Qiagen). After washing with a buffer (50 mM Tris-HCl, pH 8.0, 150 mM NaCl), the protein was eluted with a linear gradient of 20–250 mM imidazole. Fractions containing the target protein were pooled, concentrated to 0.5 mL and loaded onto a Superdex 200 increase column (GE Healthcare) equilibrated with a buffer (20 mM Tris-HCl, pH 8.0, 150 mM NaCl) for further purification.

Selenomethionine-labeled protein was expressed in M9 medium (M9 salts supplemented with 2 mM $MgSO_4$, 0.1 mM $CaCl_2$, 0.5% w/v glucose, 2 mg/L biotin, 2 mg/L thiamine, 0.03 mg/L $FeSO_4$). At an $OD_{600}$ of 0.5, 100 mg/mL of phenylalanine, lysine, and threonine, 50 mg/mL of isoleucine, leucine, and valine, as well as 80 mg/mL of selenomethionine (Chemie Brunschwig) were added as solid powder to the cultures, which were further incubated for 30 min. Expression was then induced with 0.2 mM IPTG and cells were further incubated at 16°C on a shaker for 16 hr. Cells were harvested at

5000x$g$ for 15 min, 4°C and pellets were resuspended in the lysis buffer (50 mM Tris pH 8.0, 150 mM NaCl and 5 mM β-mercaptoethanol) and was purified as described above.

## Detection of DUB activity using a suicide probe

1 µM Ub-PA and 1 µM His$_6$-Lem27 or His$_6$-Lem27$_{C24A}$ was mixed in 20 µL DUB buffer (50 mM Tris-HCl pH 7.5, 50 mM NaCl, 2 mM DTT) and incubated at 23°C for 1 hr. Reaction was terminated by 5 µL 5 × SDS loading buffer. Samples were heated at 95°C for 5 min prior to being resolved by SDS-PAGE and the proteins were detected by silver or Coomassie brilliant blue staining.

## Cleavage of diubiquitin

1.5 µM of linear diubiquitin or diubiquitin (Boston Biochem) linked by each of the seven lysine residues in ubiquitin was mixed with 1.0 µM purified proteins in 20 µL DUB buffer (500 mM Tris-HCl pH 7.5, 1 M NaCl, 10 mM DTT) and incubated at 37°C for 10 min or 2 hr. The reactions were terminated by 5 × SDS loading buffer and samples were boiled for 5 min prior to being separated by SDS-PAGE. Proteins were detected by Coomassie brilliant blue staining or by immunoblotting.

## Cell cultures and transfection

HEK293T and Hela cells were grown in Dulbecco's modified minimum Eagle's medium (DMEM) supplemented with 10% fatal bovine serum (FBS). RAW264.7 and U937 cells were grown in RPMI 1640 medium containing 10% FBS. The medium was supplemented with 100 µg/mL penicillin and 10 µg/mL streptomycin when necessary. The cells were grown at 37°C with 5% CO$_2$. U937 cells were differentiated into macrophages with phorbol-12-myristate-13-acetate (PMA) as described earlier (*Tilney et al., 2001*). All cell lines were from ATCC and were authenticated by short tandem repeat (STR) analysis and were free of mycoplasma contamination as examined by a PCR-based test (Sigma, cat# MP0025).

To determine the DUB activity in cells, a construct for expressing Flag-Ub (*Sheedlo et al., 2015*) was cotransfected into HEK293T with a plasmid expressing GFP-Lem27 or GFP-Lem27$_{C24A}$ using lipofectamine 3000 (Life Technology) per manufacturer's instructions. The DUB GFP-SdeA$_{1-200}$ from SdeA (*Sheedlo et al., 2015*) was included as a DUB control.

To establish a cell line stably expressing mCherry-Rab10, we first replaced the *gfp* gene in peGFPC1 (Clontech) with a DNA fragment coding for mCherry-Rab10 and the resulting plasmid pmCherry-Rab10 was transfected into MLE cells (ATCC) grown to 20–30% confluence in a 24-well plate. 24 hr after transfection, medium containing 800 µg/mL G418 was used to select for cells harboring integrated pmCherry-Rab10. As a control, the same medium was added to untransfected cells seeded in a 24-well plate. The selective medium containing G418 was replaced every 2–3 days and the cells were visually inspected for toxicity. After one week, cell death began to occur in untransfected samples, the concentration of G418 in medium for transfected samples was switched to 200 µg/mL and the cells were allowed to grow for 2 days. To isolate clones, cells diluted at a density of 1 cell per 100 µL were distributed in 96-well plates and wells that contained only 1 cell were identified under a fluorescence microscope. Several such clones were saved and one was expanded in petri dishes and used for subsequent experiments.

## Immunoprecipitation, antibodies, and immunoblotting

HEK293T cells were resuspended with 1 ml NP40 lysis buffer for 10 min on ice, and the lysates were then centrifuged at 12,000x$g$ at 4°C for 10 min. Beads coated with Flag- or HA-specific antibody were added to cleared lysates and incubated on a rotatory shaker for 8 hr at 4°C. After washed three times with the NP40 lysis buffer, aliquots of beads were mixed with the purified Lem27 or Lem27$_{C24A}$ in 20 µL DUB buffer at 37°C. At the indicated time points, reactions were stopped by adding 5 µL 5 × SDS loading buffer and were heated at 95°C for 5 min. After SDS-PAGE, proteins were transferred onto nitrocellulose membranes (Pall Life Sciences) for immunoblotting after being blocked in 5% nonfat milk in PBST buffer for 1 hr. Primary antibodies used in this study and their dilutions are as follows: α-Flag(Sigma, Cat# F1804, 1: 3000), α-GFP(Sigma, cat# G7781, 1:5000), α-His(Sigma, cat# H1029, 1: 10,000), α-HA (Santa Cruz, cat# sc-7392, 1: 1000), α-ICDH (1: 20,000) (*Xu et al., 2010*), α-tubulin (DSHB, E7, 1: 10,000). Antibodies specific for Lem27 were generated by immunization of rabbits with purified His$_6$-Lem27 using a standard procedure (AbMax Biotechnology

Co., LTD, Beijing, China) and were used at 1:500. Washed membranes were incubated with appropriate IRDye secondary antibodies and signals were detected and analyzed by an Odyssey CLx system (LI-COR).

## In vitro ubiquitination assays

To determine SidC-induced ubiquitination of Rab10, HEK293T cells transfected to express 4xFlag-Rab10 for 24 hr were lysed with the NP40 lysis buffer and 4xFlag-Rab10 was purified by immunoprecipitation with agarose beads coated with Flag-specific antibody as described above, bound protein was eluted with a ubiquitination buffer (*Hsu et al., 2014*) containing 100 μg/ml 3xFlag peptide (Sigma). Ubiquitination reactions were established by adding 1.0 μM purified $His_6$-SidC, 2.5 μM ubiquitin and 0.5 mM ATP in 100 μL reactions. The reactions were allowed to proceed for 2 hr at 37°C before the addition of purified 1.0 μM Lem27 or $Lem27_{C27A}$. After incubation at 37°C for an additional 2 hr. Reactions were terminated by adding 6x SDS sample buffer. After denaturing the proteins by boiling for 5 min, samples were resolved by SDS-PAGE, transferred to nitrocellulose membranes and probed with appropriate antibodies as described above.

## Bacterial infections, immunostaining and image analysis

For infection experiments, *L. pneumophila* strains grown to the post-exponential growth phase ($OD_{600}$ = 3.3–3.8) were used for infection at the indicated MOIs. For intracellular growth assays in RAW264.7 cells or in *D. discoideum*, infections were performed at an MOI of 0.05. Extracellular bacteria were removed by washing infected samples with warm PBS 2 hr after adding the bacteria. At the indicated time points, cells were lysed with 0.2% saponin and appropriate dilutions of the lysates were plated on CYE plates, bacteria colonies were counted after 5 day incubation at 37°C. To determine the impact of the Lem27 on SidC-induced Rab10 ubiquitination in infected cells, HEK293T cells transfected to express 4xFlag-Rab10 and the FcγII receptor (*Kagan and Roy, 2002*) for 24 hr were infected with the indicated bacterial strains opsonized with Legionella-specific antibodies as described earlier (*Qiu et al., 2016*).

For immunostaining experiments, macrophages differentiated from U937 cells with PMA (*Tilney et al., 2001*) were infected with relevant *L. pneumophila* strains at an MOI of 10 for 2 hr. Samples fixed with 4% paraformaldehyde were immunostained with appropriate antibodies as described earlier (*Sheedlo et al., 2015*). Briefly, fixed cells were washed three times with PBS prior to being permeabilized with 0.2% Triton X-100 at room temperature for 1 min or with cold methanol (−20°C) for 10 s. Samples were then blocked in 4% goat serum for 30 min followed by incubating with primary antibodies for 1 hr in 4% goat serum. Antibodies used are: α-*L. pneumophila* (1:10,000) (*Xu et al., 2010*), α-Flag(Sigma, cat# F1804, 1: 200) and α-FK1 (Enzo, Prod. No. BML-PW8805, 1:1,000). After 3x washing with PBS, samples were incubated with appropriate secondary antibodies conjugated to specific fluorescence dyes. Coverslips mounted on slides with nail polish were used for observation and image acquisition using an Olympus IX-83 fluorescence microscope.

To quantitate fluorescence signals, images of randomly chosen fields acquired using an Olympus IX-83 with identical parameters were analyzed using Photoshop. To determine the average signal intensity of a given fluorescence signal, at least 150 randomly chosen cells were defined and the signal intensity was determined by measuring the gray value. Average intensity per cell was calculated by dividing the sum of the gray value by the number of cells analyzed. For signal intensity of a given fluorescence signal associated with bacterial phagosomes, an identical rectangular area with a fixed aspect ratio was used to measure signal. In each case, at least 150 vacuoles were analyzed per sample. All image analyses were performed blind by coding the samples from the beginning of the experiments.

## Protein crystallization and data collection

To obtain Lem27-Ub-PA, 5 mg $Lem27_{1-417}$ was incubated with ubiquitin-propargylamide (Ub-PA) (UbiQ) at a 1:1.2 molar ratio at 4°C for 30 min. The mixture was loaded onto a Superdex 200 increase column (GE Healthcare) to purify the $Lem27_{1-417}$-Ub-PA conjugate, which was then concentrated using an Amicon Ultra 30 K centrifugal filter (4,000 g, 4°C) to approximately 18 mg/mL.

After screening Lem27 and multiple fragments of the protein for crystallization, only the $Lem27_{1-417}$-Ub-PA conjugate was found to crystallize. For crystallization, $Lem27_{1-417}$-Ub-PA was mixed with

the reservoir solution at an equal volume and crystallized by the sitting drop vapor diffusion method at 16°C. Crystals of the Lem27$_{1-417}$-Ub-PA conjugate were obtained within three days in the condition containing 100 mM Magnesium formate, 15% (w/v) PEG3,350. We then used SeMet Lem27$_{1-417}$-Ub-PA for optimization. After extensive attempts, diffraction quality crystals of Lem27$_{1-417}$-Ub-PA were grown in the presence of 100 mM Magnesium formate, 5% (w/v) PEG3,350. Crystals were harvested with 20% (v/v) ethylene glycol as a cryoprotectant before flash freezing them in liquid nitrogen.

## Structure determination and refinement

Diffraction data were collected at the Shanghai Synchrotron Radiation Facility (SSRF) BL-17U1 and a single-wavelength anomalous diffraction (SAD) dataset was obtained and the data were processed with the HKL-2000 package (*Otwinowski and Minor, 1997*). Autosol program of PHENIX package (*Adams et al., 2010*) was used for SAD phasing, followed by iterative manual building using Coot (*Emsley and Cowtan, 2004*) and refinement using PHENIX. The crystals belong to space group *P* 1 2$_1$one with unit-cell dimensions of a = 66.75 Å, b = 118.95 Å, and c = 84.28 Å. The final structure was refined at 2.43 Å resolution (Rfactor and Rfree of 21.43 and 25.52%, respective) (*Table 1*). Structure quality was analyzed during PHENIX refinements and later validated by the PDB validation server. Molecular graphics were generated using PyMol (Schrödinger, LLC).

## Data quantitation, statistical analyses

Student's *t*-test was used to compare the mean levels between two groups each with at least three independent samples.

## Acknowledgements

We thank Dr. Chittaranjan Das (Purdue University) for critical reading of the manuscript. This work was supported by research fund from the First Hospital of Jilin University, the Thousand Young Talents Program of the Chinese government (JZQ) and startup fund from Jilin University, National Natural Science Foundation of China grants (31770149 and 31970134 to JZQ; 31770948 and 31570875 to SO), Marine Economic Development Special Fund of Fujian Province (FJHJF-L-2020–2) (SO) and the High-level personnel introduction grant of Fujian Normal University (Z0210509) (SO). The diffraction data were collected at the beamline BL-17U1 of Shanghai Synchrotron Radiation Facility (SSRF).

## Additional information

### Funding

| Funder | Grant reference number | Author |
| --- | --- | --- |
| National Natural Science Foundation of China | 31770149 | Jiazhang Qiu |
| National Natural Science Foundation of China | 31970134 | Jiazhang Qiu |
| National Natural Science Foundation of China | 31770948 | Songying Ouyang |
| National Natural Science Foundation of China | 31570875 | Songying Ouyang |
| The First Hospital of Jilin University | Research Fund | Shuxin Liu |
| Chinese government program | The Thousand Young Talent | Jiazhang Qiu |
| Jilin University | Startup fund | Jiazhang Qiu |
| Fujian Province(FJHJF-L-2020–2) | Marine Economic Development Special Fund | Songying Ouyang |
| Fujian Normal University | (Z0210509) | Songying Ouyang |

The funders had no role in study design, data collection and interpretation, or the decision to submit the work for publication.

## Author contributions

Shuxin Liu, Formal analysis, Validation, Investigation, Methodology, Writing - original draft; Jiwei Luo, Data curation, Investigation; Xiangkai Zhen, Data curation, Formal analysis, Investigation; Jiazhang Qiu, Conceptualization, Resources; Songying Ouyang, Resources, Formal analysis, Supervision, Validation, Investigation, Project administration, Writing - review and editing; Zhao-Qing Luo, Conceptualization, Resources, Formal analysis, Supervision, Validation, Writing - original draft, Project administration, Writing - review and editing

## Author ORCIDs

Jiazhang Qiu  https://orcid.org/0000-0002-7723-5073
Songying Ouyang  https://orcid.org/0000-0002-1120-1524
Zhao-Qing Luo  https://orcid.org/0000-0001-8890-6621

## Decision letter and Author response

Decision letter https://doi.org/10.7554/eLife.58114.sa1
Author response https://doi.org/10.7554/eLife.58114.sa2

# Additional files

## Supplementary files

- Supplementary file 1. DALI search results against the PDB using the $Lem27_{1-417}$ structure.
- Supplementary file 2. Bacterial strains, plasmids and primers used in this study.
- Transparent reporting form

## Data availability

Diffraction data have been deposited in PDB under the accession code 7BU0.

The following dataset was generated:

| Author(s) | Year | Dataset title | Dataset URL | Database and Identifier |
|---|---|---|---|---|
| Zhen X, Luo J, Ouyang S, Liu S, Qiu J, Luo ZQ | 2020 | Crystal structure of an OTU deubiquitinase in complex with Ub-PA | https://www.rcsb.org/structure/7BU0 | RCSB Protein Data Bank, 7BU0 |

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
