## [Decision Letter]

**Acceptance summary:**

This paper describes an analysis of Lem27, a distant related OTU-like DUB encoded by legionella. The authors demonstrate that Lem27 cleaves multiple ubiquitin chain linkage types but prefers K6, K11, and K48. They also solve the structure of Lem27-Ub, revealing both a distantly related OTU fold as well as unique features. They identify residues important for Ub binding and show that some of these affect cleavage rates. Lem27 was found to associate with the Legionelle vacuole in cells and to remove ubiquitin from this structure. The further linked this to Rab10 ubiquitylation, which is linked with SidC activity in cells.

**Decision letter after peer review:**

Thank you for submitting your article "Interplay between bacterial deubiquitinase and ubiquitin E3 ligase regulates ubiquitin dynamics on Legionella phagosomes" for consideration by *eLife*. Your article has been reviewed by three peer reviewers, one of whom is a member of our Board of Reviewing Editors, and the evaluation has been overseen by Cynthia Wolberger as the Senior Editor. The following individuals involved in review of your submission have agreed to reveal their identity: Yuxin Mao (Reviewer #2); Elizabeth Hartland (Reviewer #3).

The reviewers have discussed the reviews with one another and the Reviewing Editor has drafted this decision to help you prepare a revised submission.

As the editors have judged that your manuscript is of interest, but as described below that additional experiments are required before it is published. We would like to draw your attention to changes in our revision policy that we have made in response to COVID-19 (https://elifesciences.org/articles/57162). First, because many researchers have temporarily lost access to the labs, we will give authors as much time as they need to submit revised manuscripts. We are also offering, if you choose, to post the manuscript to bioRxiv (if it is not already there) along with this decision letter and a formal designation that the manuscript is "in revision at *eLife*". Please let us know if you would like to pursue this option. (If your work is more suitable for medRxiv, you will need to post the preprint yourself, as the mechanisms for us to do so are still in development.)

Summary:

This paper describes an analysis of Lem27, a distant related OTU-like DUB encoded by legionella. The authors demonstrate that Lem27 cleaves multiple ubiquitin chain linkage types but prefers K6, K11, and K48. They also solve the structure of Lem27-Ub, revealing both a distantly related OTU fold as well as unique features. The identify residues important for Ub binding and show that some of these affect cleavage rates. Lem27 was found to associate with the Legionelle vacuole in cells and to remove ubiquitin from this structure. The further linked this to Rab10 ubiquitylation, which is linked with SidC activity in cells.

Essential revisions:

The reviewers felt that 2 aspects of the paper need to be addressed in detail.

First, In the experiment in Figure 4C, an HA-tagged single lys ubiquitin mutant was expressed in cells and the enriched materials from HA-IP was treated with recombinant Lem27. The authors claimed that Lem27 prefers to cleave K6, K11, and K48 chains based on the disappearance of poly-ub signal. However, there is a major caveat in the design of the experiment and the interpretation of the results. Ub mutants carrying a single lys do not necessarily form one type of Ub linkage with their remaining single lys residue; instead these Ub mutants can be present in any type of ub chains since they can be attached to endogenously wild-type Ub as a donor ub. Thus the interpretation of the results from Figure 4C is not valid. This may also explain why Lem27 cleaves ubiquitin chains from samples prepared from Ub_K27_ (subsection “Lem27 preferentially cleaves diubiquitin linked by K6, K11 or K48”).

Also, SidC has been previously shown to catalyze both K11, K33, and K63 ub chains and to a less extend K48 chains. In Figure 7A (left panel), the ubiquitinated flag-Rab10-ub is likely just mono-ubiquitination, this experiment does not justify that Lem27 prefers cleave K11 linkage. The way it is written in the text, (subsection “Lem27 and the SidC family of E3 ubiquitin ligases function in concert to regulate protein ubiquitination on bacterial phagosomes”) sounds like SidC/SdcA only catalyze K11 chain and Lem27 specifically cleave K11 chain.

So both of these aspects need to be addressed in the text. We also suggest that Figure 4C be moved to the supplement.

Second, there are a number of issues with the functional assays in cells, which will need to be address through new experiments.

i) The localisation of Lem27 is shown at only one timepoint (Figure 3). What happens to the localisation of Lem27 over a time course of infection, say up to 18 h? This could indicate that Lem27 has other roles/targets in the cell.

ii) Is Lem27 also associated with the LCV in amoebae and is the mutant attenuated in amoebae?

iii) You state that "similar results were obtained in three independent experiments". All these independent experiments should be combined and statistical analysis done on the pooled data, not on a single experiment encompassing 100 vacuoles. Importantly, vacuole counting of samples must be performed in a blinded manner.

iv) Figure 7B should also show counting of vacuoles that are positive for Rab10 for each strain (again done in a blinded manner).

v) It is understood that SidC/Rab10 was used as a proof in principle that Lem27 can reverse K11 ubiquitination. Still it would be good to know how often Lem27 localisation overlaps with SidC and Rab10 (and to compare localisation of native Lem27 and the catalytic mutant). Again, this points to the possibility of other targets if Lem27 is not always/often associated with SidC/Rab10.

---

## [Author Response]

Essential revisions:The reviewers felt that 2 aspects of the paper need to be addressed in detail.First, In the experiment in Figure 4C, an HA-tagged single lys ubiquitin mutant was expressed in cells and the enriched materials from HA-IP was treated with recombinant Lem27. The authors claimed that Lem27 prefers to cleave K6, K11, and K48 chains based on the disappearance of poly-ub signal. However, there is a major caveat in the design of the experiment and the interpretation of the results. Ub mutants carrying a single lys do not necessarily form one type of Ub linkage with their remaining single lys residue; instead these Ub mutants can be present in any type of ub chains since they can be attached to endogenously wild-type Ub as a donor ub. Thus the interpretation of the results from Figure 4C is not valid. This may also explain why Lem27 cleaves ubiquitin chains from samples prepared from Ub_K27_ (subsection “Lem27 preferentially cleaves diubiquitin linked by K6, K11 or K48”).

We’d like to thank the reviewer for this insightful comment. We agree that the polyubiquitin chains produced in cells overexpressing the Ub mutants are not necessarily uniformly linked via the single K remaining in the mutants due to the presence of endogenous ubiquitin. We have followed the suggestion to remove Figure 4C from the manuscript.

Also, SidC has been previously shown to catalyze both K11, K33, and K63 ub chains and to a less extend K48 chains. In Figure 7A (left panel), the ubiquitinated flag-Rab10-ub is likely just mono-ubiquitination, this experiment does not justify that Lem27 prefers cleave K11 linkage. The way it is written in the text, (subsection “Lem27 and the SidC family of E3 ubiquitin ligases function in concert to regulate protein ubiquitination on bacterial phagosomes”) sounds like SidC/SdcA only catalyze K11 chain and Lem27 specifically cleave K11 chain.So both of these aspects need to be addressed in the text. We also suggest that Figure 4C be moved to the supplement.

We agreed with this criticism and have revised the text accordingly. Please see the subsection “Lem27 preferentially cleaves diubiquitin linked by K6, K11 or K48”.

Second, there are a number of issues with the functional assays in cells, which will need to be address through new experiments.i) The localisation of Lem27 is shown at only one timepoint (Figure 3). What happens to the localisation of Lem27 over a time course of infection, say up to 18 h? This could indicate that Lem27 has other roles/targets in the cell.

We understand the importance of knowing the cellular localization of an effector at different phases of infection. Please note that here the association of Lem27 with the LCV was determined by using a Flag-specific antibody to stain 4xFlag-Lem27 expressed from a plasmid. Because the association with the LCV by a given effector is controlled by its abundancy, time of expression during infection and other factors, which are affected by factors such as protein stability and expression in response to specific cues, meaningful data on the kinetics of LCV association should be determined with antibodies specific for the effectors itself (i.e. determination of the kinetics of the association of Lem27 with the LCV by ectopically protein may not reflect the kinetics of endogenous protein).

Unfortunately, the polyclonal antibodies against Lem27 we prepared are not suitable for immunostaining. The antibodies nonspecifically recognize unknown proteins in fixed cells, leading to high background and making the signals for Lem27 undiscernible.

We have attempted to circumvent the problem by introducing 1xFlag-*lem27* into the *lem27* locus on the *L. pneumophila* chromosome by a knock-in procedure using strain Lp02∆*lem27* as the recipient. Although the knock-in experiment worked and we successfully obtained a *L. pneumophila* strain (we called it Lp02(cFlag-Lem27)) that expresses 1xFlag-Lem27 from its cognate promoter on the chromosome, the protein can be detected by the Lem-27-specific antibodies but not the Flag antibody. As a result, we were unable to detect the association of 1xFlag-Lem27 with the LCV by immunostaining using the Flag antibody in infections using strain Lp02(cFlag-Lem27).

**Author response image 1. sa2fig1:** Flag-tagged Lem27 expressed from a knockin *L. pneumophila* strain can be detected by Lem27 antibodies but not by Flag antibody. A construct harboring Flag-*lem27* on a suicide plasmid was used to introduce tagged *lem27* into the ∆*lem27* strain make the knockin strain in which Flag-Lem27 will be expressed from its cognate promoter. A. Bacteria harboring Flag-*lem27* were identified by colony PCR. The two strains harboring knocked in gene were marked by “+”. B. Expression of Lem27 or Flag-Lem27 in relevant *L. pneumophila* strains. Total protein lysates of the indicated strains resolved by SDS-PAGE were detected by our Lem27-specific antibodies (top panel). Note the presence of a band in all strains expressing Lem27, including strain cFlag-Lem27, with the exception of strain ∆*lem27* (5^th^ lane). Flag-Lem27 was not detected by the Flag antibody (lower panel).

This technical barrier imposed by the lack of a key reagent (high quality Lem27-specific antibodies suitable for immunostaining) also prevented us from analyzing the association of Lem27 with the LCV at later phase of infection.

In sum, despite considerable efforts, we were unable to determine the kinetics of LCV association by endogenous Lem27. The kinetics of LCV association and the potential existence of additional cellular targets of Lem27 will be addressed at least in part by the development of better antibodies and effective methods in the identification of DUB substrates, which is a focus of ongoing study in our laboratory.

ii) Is Lem27 also associated with the LCV in amoebae and is the mutant attenuated in amoebae?

We have performed infections with the amoebae host *Dictyostelium discoideum* with the bacterial strains used in BMDMs. Our results indicate that Lem27 is associated with the LCV in a way similar to what we observed in experiments using mammalian cells. We have included these results in Figure 3—figure supplement 1.

We have also examined the ∆*lem27* mutant for intracellular replication in *D. discoideum* and found that this mutant did not display detectable defects in this host (Figure 3—figure supplement 3).

iii) You state that "similar results were obtained in three independent experiments". All these independent experiments should be combined and statistical analysis done on the pooled data, not on a single experiment encompassing 100 vacuoles. Importantly, vacuole counting of samples must be performed in a blinded manner.

We agreed with this suggestion and have revised the results by using data from three independent experiments. Also, these experiments were performed in blind (the bacterial strains used were coded by lab member and the experiments (infection, staining and counting) were performed by another member without knowing the identity of these strains used for the experiments). When applicable, the revised figure now reflects pooled data from three independent experiments.

iv) Figure 7B should also show counting of vacuoles that are positive for Rab10 for each strain (again done in a blinded manner).

We have added the results of Rab10-positive vacuoles for each strains done in blind (Figure 7E).

v) It is understood that SidC/Rab10 was used as a proof in principle that Lem27 can reverse K11 ubiquitination. Still it would be good to know how often Lem27 localisation overlaps with SidC and Rab10 (and to compare localisation of native Lem27 and the catalytic mutant). Again, this points to the possibility of other targets if Lem27 is not always/often associated with SidC/Rab10.

We agree that it is nice to have this piece of information. Yet, limited by the number of fluorescence colors that can be used in the same samples for immunostaining, we were unable to distinctly label SidC, Lem27 and Rab10 in the same infected cells.

Instead, we have stained the vacuoles for Rab10 and SidC and for Lem27 and SidC, respectively. Our results show that at 2 h post infection, about 70% of the vacuoles are positive for both SidC and Rab10, and approximately 34% of the vacuoles are positive for both SidC and Lem27. These results do not argue against the notion that Lem27 targets other proteins in infected cells, a module we actually support. Yet, we are restrained from over interpreting these staining data because Lem27 was expressed from a multi-copy plasmid. The existence of other ubiquitinated proteins as targets of Lem27 is a topic of our investigation, which we wish to present in future publications in conjunction with other Legionella DUBs that share substrate preference with Lem27 in in vitro biochemical assays.